# Numerical simulations of ice accretion on wind turbine blades: are performance losses due to ice shape or surface roughness?

Francesco Caccia and Alberto Guardone

Department of Aerospace Science and Technology, Politecnico di Milano, Via La Masa 34, 20156 Milan, Italy.

**Correspondence:** Francesco Caccia (francescoangelo.caccia@polimi.it)

**Abstract.** Ice accretion on wind turbine blades causes both a change in the shape of its sections and an increase in surface roughness. These lead to degraded aerodynamic performances and lower power output. Here, a high-fidelity multi-step method is presented and applied to simulate a 3-hour rime icing event on the NREL 5 MW wind turbine blade. Five sections belonging to the outer half of the blade were considered. Independent time steps were applied to each blade section to obtain detailed ice shapes. The roughness effect on airfoil performance was included in CFD simulations using an equivalent sand-grain approach. The aerodynamic coefficients of the iced sections were computed considering two different roughness heights and extensions along the blade surface. The power curve before and after the icing event was computed according to the Design Load Case 1.1 of the International Electrotechnical Commission. In the icing event under analysis, the decrease in power output strongly depended on wind speed and, in fact, tip-speed ratio. Regarding the different roughness heights and extensions along the blade, power losses were qualitatively similar but significantly different in magnitude, despite the well-developed ice shapes. It was found that extended roughness regions in the chordwise direction of the blade can become as detrimental as the ice shape itself.

## 1 Introduction

Arguably, wind energy will lead the energy transition in Europe. In 2022, it produced 15% of the total generated electricity (Jones et al., 2023). Its share is due to increase to approximately 30% by 2030 and 45% by 2050 to reach carbon neutrality (European Commission, 2018). Future installations of wind turbines will mainly occur in cold regions. Higher wind speeds and air density guarantee a higher wind power density, leading to a competitive cost of energy and viable investments (Directorate-General for Energy, 2012). However, two phenomena may reduce the power output in cold climates when considering a single wind turbine or a wind farm. First, cold climates often lead to a stable atmospheric boundary layer, causing less mixing of the wake of a wind turbine. Thus, wake effects are stronger, and the wind turbines located downwind in wind farms produce less power. Second, ice can form on wind turbines when clouds or super-cooled fog arise at low elevations, and temperatures drop below 0 °C. These conditions may persist for days or even weeks and, due to climate change, they may occur in previously unexpected locations. In February 2021, three consecutive winter storms swept over Texas and caused a more than 80% reduction in energy output compared to the previous week. It is shown in Fig. 1.

Indeed, ice may affect wind turbines in several ways. The first visible effect is the blade aerodynamics degradation, reducing the wind turbine power output. Instrumentation and controller errors may follow. As more ice is accreted, the structural

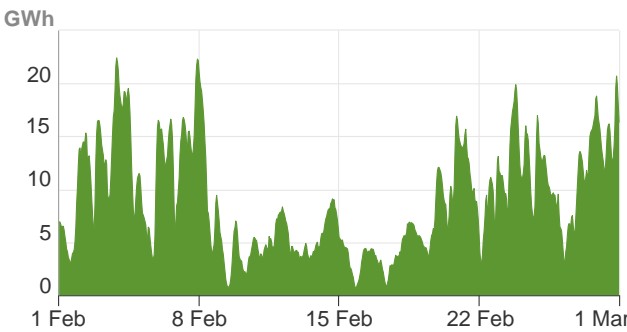

**Figure 1.** Texas region electricity generation by wind energy in February 2021. Winter storms occurred between February 10 and February 20. Source: U.S. Energy Information Administration, Hourly Electric Grid Monitor. Available at: https://www.eia.gov/electricity/gridmonitor (Accessed: 2022-01-10).

behaviour changes as well and the fatigue life of the structure can be affected. Ice shedding may also be a severe threat, endangering equipment and people nearby and causing significant load unbalances on the rotor. In many cases, a shutdown may become unavoidable. For these reasons, wind turbines operating in cold regions must be equipped with an Ice Protection System (IPS). Electro-thermal IPSs provide a possible solution. These devices are energy-consuming, especially if run in anti-icing
mode (Caccia et al., 2022). Recent design solutions combine the effects of centrifugal force and IPS heat to remove ice from the blades (Getz and Palacios, 2021). Better predictions of power losses due to icing may lead to improved designs and further energy savings during the operation of such devices.

The problem of ice accretion on wind turbines has been long studied but is still of great interest. In 2002, the International Energy Agency (IEA) established a cooperation group of international experts, Task 19, to study wind energy in cold climates.
The working group has been running ever since. A recent report by the Task 19 group reviewed the technologies available for wind turbines operating in cold climates (Lehtomäki et al., 2018). Many fundamental aspects were covered, including ice detection systems, ice protection systems, ice accretion models, ice shedding, operation and maintenance, standards, and testing. Ice accretion models were categorised into two classes, i.e., simplified and advanced. Simplified models are typically used within weather models to estimate icing rate and mass on objects. They are based on empirical relationships to simulate ice growth
on cylinders. On the other hand, advanced models are used for aerodynamic analyses by capturing detailed physics of the ice accretion process to produce accurate 2D or 3D ice shapes. State-of-the-art advanced ice accretion models were benchmarked in 2022 at the 1st AIAA Ice Prediction Workshop for code-to-code and code-to-experiment comparisons, including Fensap-Ice (Ozcer et al., 2022), GlennICE (Wright et al., 2022), IGLOO3D (Radenac and Duchayne, 2022), and PoliMIce (Morelli et al., 2022). A review of the results was written by Laurendeau et al. (2022). In this work, we analysed the aerodynamics of iced
blade sections in detail. So, an advanced ice accretion model was used with the in-flight ice accretion engine PoliMIce (Gori et al., 2015).

Depending on atmospheric conditions, different types of ice can form. Their standard classification is *rime*, *glaze*, and *mixed* ice. The current study focused on rime ice, formed when super-cooled water droplets freeze instantly upon impact. The type of

ice and the accretion rate depend on various parameters. According to Etemaddar et al. (2014), such parameters can be divided into two categories, i.e., atmospheric and system parameters. The atmospheric parameters are: the free stream Temperature $T_\infty$; the Liquid Water Content LWC, defining the amount of water dispersed in a reference volume of air; and the Median Volumetric Diameter MVD, defining the droplet diameter above and below which half the volume of water is contained. The system parameters are: the geometric parameters of the object, i.e., shape, orientation, and dimension (shape, angle of attack, chord $c$, and thickness $t$ of an airfoil); and the relative wind speed $V_{\mathrm{rel}}$. The formation of rime ice is favoured by low $T_\infty$, low LWC, small MVD and low $V_{\mathrm{rel}}$. Moreover, the rate of ice accretion increases with increasing LWC, MVD, $V_{\mathrm{rel}}$ and decreasing $t/c$. As a result, the ice mass accreted on a wind turbine blade increases from the root to the tip. For a more in-depth analysis, the reader can refer to the works by Etemaddar et al. (2014), Homola et al. (2010a, b), and Virk et al. (2010).

When ice is formed on an airfoil, two main factors alter its performance: the ice shape and the increase in surface roughness. While the former can be assessed numerically, the latter needs to be estimated either experimentally or using empirical correlations. During ice accretion, the roughness height evolves in a complex way with both space and time (Steiner and Bansmer, 2016; McClain et al., 2017, 2020). In CFD simulations, it is common practice to map the actual roughness distribution into the so-called equivalent sand grain roughness Nikuradse (1950). For this type of roughness, the shift of the velocity profile in the logarithmic region of the boundary layer is known as a function of the roughness height, $k_s$, in wall units ($k_s^+$). This topic will be addressed later.

Several numerical studies on power losses due to ice accretion are available in the literature. An exhaustive review was carried out by Contreras Montoya et al. (2022). The procedure for estimating power losses usually relies on the Blade Element Momentum (BEM) theory. It can be summarised as follows: (1) computation of the aerodynamic coefficients of the clean airfoils; (2) simulation of steady-state ice accretion on relevant 2D sections; (3) computation of the aerodynamic coefficients of the iced airfoils; and (4) computation of the power curves. Indeed, the numerical study of 2D sections can provide an affordable and reliable approximation of 3D ice accretion. In terms of ice shape, when ice accretion is coupled with the BEM solution, the two approaches are almost equivalent (Switchenko et al., 2014). In terms of airfoil performance, an experimental and numerical study on the 3D scan of an iced wind turbine airfoil was carried out by Knobbe-Eschen et al. (2019). The authors found that accurate predictions of the numerical 3D solution can be obtained by studying 2D slices of the actual ice shape, i.e., including the localised macroscopic roughness. On the other hand, the airfoil performance was over-predicted when the ice shape was span-wise averaged, i.e., macroscopic and microscopic roughness were removed. In this context of 2D BEM simulations of ice accretion, three representative works on the NREL 5 MW Reference Wind Turbine (Jonkman et al., 2009) are reported here.

One of the first efforts to study ice accretion numerically on a wind turbine was carried out by Homola et al. (2012). Five sections belonging to the whole blade span were studied. A BEM code was used to evaluate both boundary conditions for each section and wind turbine performances. The wind turbine was operating with $V_\infty = 10\,\mathrm{ms}^{-1}$, $T = -10\,°\mathrm{C}$, LWC $= 0.22$ $\mathrm{gm}^{-3}$, and MVD $= 20\,\mathrm{\mu m}$ for a total time of 1 h. Roughness was applied on the entire blade surface. The equivalent sand-grain roughness height $k_s/c$ was not specified. The power loss was about 25% between $7\,\mathrm{ms}^{-1}$ and $11\,\mathrm{ms}^{-1}$ in a steady wind.

In 2013, Turkia et al. studied power losses on a down-scaled version of the NREL 5 MW to achieve a rated power of 3 MW. Two sections belonging to the outer third of the blade were studied. The considered atmospheric conditions were $V_\infty = 7$

ms$^{-1}$, $T_\infty = 7$ °C, LWC $= 0.2$ gm$^{-3}$, and MVD $= 25$ μm for a total time of 10 h. Roughness was included in two different ways. A large-scale roughness was applied to the predicted ice shapes, while the effect of small-scale roughness was included by correcting the drag coefficients of the iced airfoils with a method proposed by Bragg (1982) for rime ice. The relation depends on equivalent sand-grain roughness height $k_s/c$, estimated between $0.9 \cdot 10^{-3}$ and $1 \cdot 10^{-3}$ using Shin's relation (Shin et al., 1991). However, Bragg's relation was developed to estimate the drag coefficient of an iced airfoil in the presence of leading edge roughness using the clean $C_D$ as input and not the iced one. As a result, the effect of drag on each section was considered twice. At the end of the icing event, power losses were approximately 25% in the 6 ms$^{-1}-$ 12 ms$^{-1}$ range and occurred up to 17 ms$^{-1}$. No information was provided on the type of wind used as input.

Finally, Etemaddar et al. (2014) simulated a 24-hour icing event on the NREL 5 MW, varying the atmospheric conditions in the considered time window and sampling them every 15 min. Six sections in the outer half of the blade were considered. Ice accretion was simulated with LEWICE (Wright, 2008). Roughness was applied to the first 25% of the airfoil chord, and $k_s/c$ was set to $0.5 \cdot 10^{-3}$. Wind turbine operation was simulated with HAWC2 (Larsen and Hansen, 2007) with the Mann spectral tensor model for atmospheric turbulence (Mann, 1994). In this case, power losses were about 45% at cut-in wind speed, reducing to 34%, 23%, and 1.8% at 7 ms$^{-1}$, 11 ms$^{-1}$, and 16 ms$^{-1}$, respectively. Icing loads were studied as well.

These results do not show a clear trend in power losses. At 11 ms$^{-1}$, the reduction in extracted power was comparable regardless of the duration of the icing event. Moreover, the trend found for increasing wind speeds was different. Homola et al. and Turkia et al. predicted a slight increase in power loss from 7 ms$^{-1}$ to 11 ms$^{-1}$, while Etemaddar et al. found a drastic decrease. Indeed, different atmospheric conditions led to different ice shapes, resulting in different aerodynamic performances of the airfoils. However, in each work, the roughness was taken into account profoundly differently, which may have led to this results pattern.

A study by Switchenko et al. (2014) supports this hypothesis. In their work, the authors numerically simulated a real-world icing event on a 1.5 MW wind turbine. Two important conclusions were drawn from this study. The first one, as already mentioned, was that simulating ice accretion on 2D sections with the BEM methodology provides very similar results to the 3D solution. The second one was that "roughness of the ice can at times be more significant than the actual size, shape and placement of the accreted ice", and so "more research is needed to evaluate the effect of atmospheric icing on wind turbine blades and their surface roughness characteristics". However, the authors never computed the airfoil polars since the BEM method was integrated within iterative CFD simulations. Thus, the nature of power losses remained unknown. On the other hand, the detrimental effect of roughness on the aerodynamic coefficients of a wind turbine airfoil was shown by Blasco et al. (2017) through ice accretion experiments and wind tunnel measurements. Ice accretion time was set to 45 min to obtain streamlined ice shapes. The maximum lift decrease and drag increase at an operational angle of attack was about 25% and 220%, respectively. However, these results are different from those by Switchenko et al., in which the effect of roughness was highlighted for long-lasting icing events.

In view of the above, the aim of this work is to: (1) perform a high-fidelity ice accretion simulation on the NREL 5 MW wind turbine blade; (2) compute the aerodynamic performances of the iced blade sections as a function of roughness; and (3) assess the effect of icing in operating conditions. The icing event was long enough for streamlined, protruded ice shapes to form,

**Table 1.** Airfoils composing the wind turbine blade.

| Airfoil | Identifier | $r/R$ [%] | $t/c$ [%] | $Re_{\text{exp}}$ [−] |
|---|---|---|---|---|
| DU 99-W-405LM | DU40 | 18.7 | 40.5 | $7 \cdot 10^6$ |
| DU 99-W-350LM | DU35 | $25.2 < r/R < 31.7$ | 35.0 | $7 \cdot 10^6$ |
| DU 97-W-300LM | DU30 | 38.2 | 30.0 | $7 \cdot 10^6$ |
| DU 91-W2-250LM | DU25 | $44.7 < r/R < 51.2$ | 25.0 | $7 \cdot 10^6$ |
| DU 93-W-210LM | DU21 | $57.7 < r/R < 64.2$ | 21.0 | $7 \cdot 10^6$ |
| NACA 64$_3$-618 | NA18 | $70.7 < r/R < 100$ | 18.0 | $6 \cdot 10^6$ |

to combine the effects of ice shapes and roughness. The work was carried out using both open-source software and in-house
codes. We presented a preliminary work in this context in Caccia et al. (2021). Original contributions to the state-of-the-art
include the introduction of span-dependent time stepping in the ice accretion simulation and an analysis of the sensitivity of
the solution to roughness height and extension. It will be shown that roughness can significantly affect airfoil aerodynamics
and power production also when complex ice shapes are present.

The paper is structured as follows. The methodology is presented in Section 2, together with the setup of the numerical
simulations. In Section 3, the numerical setup is compared with experimental data. The aerodynamic coefficients of the clean
airfoils and an icing experiment on a rotating blade section were reproduced. In Section 4, the results of ice accretion are
presented. Then, the aerodynamic coefficients of the iced sections are computed and used to simulate the power curve of the
wind turbine. Finally, conclusions are drawn in Section 5.

## 2 Methodology

The NREL 5 MW reference wind turbine was analysed in this work. The blade structural and aerodynamic designs are based
on the Dutch Offshore Wind Energy Converter (DOWEC) project (Kooijman et al., 2003). According to a more detailed design
of the blade (Resor, 2013), it is an IEC Class I and Category B wind turbine.

The aero-servo-elastic response of the wind turbine was modelled with OpenFAST[1]. Wind turbine aerodynamics was mod-
elled through the Blade Element Momentum Theory. Thus, 2D, independent sections were analysed throughout the whole
work. Within the BEM framework, the blade was discretised with 19 nodes along the blade span. It was made of two cylin-
drical sections, five DU airfoils, and one NACA airfoil. The aerodynamic coefficients of DU airfoils were measured by Ruud
van Rooij of Delft University of Technology at a Reynolds number of 7 million. NACA 64$_3$-618 coefficients were taken from
Abbott et al. (1945) at a Reynolds number of 6 million. 2D data of the airfoils are provided in the DOWEC report. A review of
the airfoils distribution along the blade, their thickness, and the tested Reynolds number is presented in Table 1.

---

[1]Available at https://github.com/OpenFAST/openfast. Accessed: 2020-09-20.

**Table 2.** Atmospheric conditions during the icing event studied.

| Duration | $V_\infty$ | Wind Shear Exponent | $P_\infty$ | $\rho_{\mathrm{air}}$ | $T_\infty$ | LWC | MVD |
|---|---|---|---|---|---|---|---|
| [min] | [ms$^{-1}$] | [$-$] | [Pa] | [kgm$^{-3}$] | [°C] | [gm$^{-3}$] | [μm] |
| 180 | 10 | 0.15 | 101325 | 1.341 | -10 | 0.22 | 20 |

The first step involved reproducing these experimental data with a CFD solver, SU2 (Economon et al., 2015). This permitted the validation of the solver setup. Reynolds-Averaged Navier-Stokes (RANS) equations were solved. Then, the numerical data were extrapolated to the entire 360° range of angles of attack (AoA) and corrected for 3D effects using NREL's tool *AirfoilPrep* to prepare them for the BEM model of the wind turbine. In particular, the Viterna Method (Viterna and Janetzke, 1982) was used for extrapolation, considering an aspect ratio of 17. The 3D corrections by Du and Selig (1998) with Eggers $C_D$ adjustment

(Eggers et al., 2003) were applied only for positive angles of attack and considering the rated rotational velocity as input, being consistent with NREL's aerodynamic data of the wind turbine model (Jonkman et al., 2009).

     The second step was the simulation of the icing event. The atmospheric conditions of the icing event on the wind turbine are reported in Table 2. The same conditions were studied by Homola et al. (2012) for 1 h and by Zanon et al. (2018) for approximately 8 h. In the current contribution, the icing event lasted for 3 h. A wind shear exponent of 0.15 was also considered

so that, according to the Normal Wind Profile model of the DNV-GL Guidelines (Germanischer Lloyd, 2010), the average horizontal wind speed $V$ as a function of the height above the ground $z$ is:

$$V(z) = V_{\mathrm{hub}}(z/z_{\mathrm{hub}})^{0.15} \tag{1}$$

where $V_{\mathrm{hub}}$ is the wind speed $V$ at the hub height $z_{\mathrm{hub}}$. These atmospheric conditions led to the formation of rime ice on the blade surface. During the icing event, wind turbine operation was computed using OpenFAST to find the equilibrium condition

of the whole system, considering wind shear and blade deformability. A steady wind was assumed at this stage. A retroaction on the wind turbine operating state due to a change in the blade aerodynamics was included. However, such retroaction would have been influential on a longer time scale or considering a more penalising roughness during ice accretion. Local boundary conditions were evaluated on significant blade sections and used as input for ice accretion. A multi-step approach was adopted, dividing the total accretion time into sub-intervals. For each section, time steps were set independently according to the required

time discretisation, taking full advantage of the BEM approximation. The ice accretion engine PoliMIce (Gori et al., 2015) was used for these simulations. The software uses SU2 for computing the aerodynamic field and PoliDrop (Bellosta et al., 2019) for Lagrangian particle tracking. The numerical setup chosen for ice accretion was validated against nine experimental test cases on a rotating section by Han et al. (2012).

     The third step was the computation of the aerodynamic coefficients of the iced sections with SU2. Roughness was modelled

using an equivalent sand-grain approach. Two values of $k_s$ were compared. Moreover, the effect of the extension of the rough region along the surface of each iced section was assessed. Once more, the numerical data were extrapolated to the 360° AoA range (Viterna and Janetzke, 1982) and corrected for 3D effects (Du and Selig, 1998; Eggers et al., 2003).

The $C_P$-TSR curves were then computed using *AeroDyn*, i.e., the aerodynamic module of OpenFAST, using the blade sectional aerodynamic coefficients and the blade geometry. The power coefficient $C_P$ is the ratio between the power extracted by the wind turbine and a conventional freestream available power:

$$C_P = \frac{P}{\frac{1}{2}\rho(\pi R^2)V_\infty^3} \tag{2}$$

where $R$ is the rotor radius and $V_\infty$ the freestream velocity. By neglecting blade elasticity, it can be shown that the quantity depends only on two system parameters, i.e., the blade pitch angle and the tip-speed-ratio TSR $= V_{\text{tip}}/V_\infty$, where $V_{\text{tip}}$ is the blade tip velocity. At a certain $V_\infty$, the controller makes the wind turbine work at a specific TSR and with a specified blade pitch. Thus, the $C_P$-TSR curve can provide valuable insight into power production and power losses of wind turbines.

The final step was the computation of the power curve of the wind turbine before and after the icing event with OpenFAST. Both steady and turbulent winds were considered. Atmospheric turbulence was modelled as defined by the International Electrotechnical Commission (IEC) for the Design Load Case (DLC) 1.1 of a Category B wind turbine (IEC 61400-1 Ed. 3, 2005) using the IEC Kaimal spectral model. The realisations of the turbulent wind were generated with TurbSim (Jonkman, 2009), considering a reference turbulence intensity of 0.16. A Weibull-averaged power was computed to provide a single figure of the severity of the icing event under the two sets of roughness under analysis. This was simply computed as:

$$P_W = \int_{V_{\text{in}}}^{V_{\text{out}}} P(V) f_w(V)\, dV \tag{3}$$

where $V_{\text{in}}$ is the cut-in wind speed, $V_{\text{out}}$ the cut-out wind speed, $P(V)$ the power curve, and $f_w(V)$ the Weibull probability density function. The latter is defined as:

$$f_w(V) = \frac{k}{c}\left(\frac{V}{C}\right)^{k-1} \exp\left[-\left(\frac{V}{C}\right)^k\right] \tag{4}$$

where $k$ is a shape parameter and $C$ is a scale parameter. The shape parameter $k$ was set to 2, to match the Rayleigh distribution advised in DNV-GL guidelines for wind turbine certification (Germanischer Lloyd, 2010). The scale parameter $C$ was set to $11.2838\ \text{ms}^{-1}$, so that the average wind speed is $V_{\text{ave}} = 10\ \text{ms}^{-1}$, as prescribed for a Class I wind turbine. The two quantities are linked by the relation

$$V_{\text{ave}}(C,k) = C \int_0^\infty e^{-t} t^{1/k}\, dt \tag{5}$$

where the integral term is the Gamma function $\Gamma(x) = \int_0^\infty e^{-t} t^{x-1} dt$ evaluated in $x = (1 + 1/k)$.

## 2.1 Law of the Wall with Roughness

Two main factors change the performance of an iced airfoil: the modification of the airfoil shape and the increase in surface roughness. While the former is computed numerically, the latter is usually estimated with empirical correlations. During ice

accretion, the roughness height evolves in a complex way with both time and space (Steiner and Bansmer, 2016; McClain et al., 2017, 2020). It is common practice in CFD simulations to map the real roughness distribution into the ideal roughness studied by Nikuradse (1950), known as equivalent sand-grain roughness. For this type of roughness, the shift in velocity profile in the logarithmic region of the boundary layer is known as a function of the roughness height $k_s$ in wall units ($k_s^+$). The following relation holds:

$$u^+ = \frac{1}{\kappa} \log\left(\frac{y^+}{k_s^+}\right) + B(k_s^+) \tag{6}$$


where $u^+$ is the non-dimensional tangential velocity in wall units, $y^+$ is the non-dimensional wall distance in wall units, $\kappa$ is the Von Kármán constant ($\kappa \approx 0.41$), and $B(k_s^+)$ is an additive constant. In particular, $y^+$, $k_s^+$, and $u^+$ are defined as:

$$y^+ = \frac{y}{\delta_\nu} \qquad\qquad k_s^+ = \frac{k_s}{\delta_\nu} \qquad\qquad u^+ = \frac{u}{u_\tau} \tag{7}$$

where $\delta_\nu = \frac{\nu}{u_\tau}$ is the viscous length scale, $u_\tau = \sqrt{\frac{\tau_w}{\rho}}$ is the friction velocity (the velocity scale of the turbulent fluctuations at
the wall), $\nu$ is the dynamic viscosity, $\tau_w$ the wall shear stress, and $\rho$ the fluid density.

Depending on the value of $k_s^+$, different roughness regimes are defined. The *smooth regime* is defined for $k_s^+ < 5$. In this case, the roughness elements are submerged within the viscous sublayer ($y^+ < 5$) and its effect on the flow is negligible. In the viscous sublayer the relation $u^+ = y^+$ holds. For $y^+ \gtrsim 30$, the clean law of the wall is recovered:

$$u^+ = \frac{1}{\kappa} \log\left(y^+\right) + 5.1 \tag{8}$$

In the *transitionally-rough* regime ($5 < k_s^+ \lesssim 70$), the additive constant $B(k_s^+)$ (Eq. 6) depends on $k_s^+$. In the *fully-rough* regime ($k_s^+ \gtrsim 70$), typical of ice, $B(k_s^+)$ becomes independent of $k_s^+$. Its value is equal to $\sim 8.0$, according to Schlichting and Gersten (2017). However, the estimation of a single $k_s$ value from a time- and space-dependent roughness distribution is not trivial. Empirical relations have been developed in the aeronautic industry. Adopting them for wind turbines is common, even if icing occurs in very different environmental conditions.

Although the use of the relation for $k_s$ developed by Shin et al. (1991) for the LEWICE code is still widespread, the code now implements a newer relation by Wright (2008). Shin's relation was specifically developed to match the ice shapes predicted by the LEWICE code to experimental ones. For this reason, it may lack of generality. On the other hand, Wright's relation "was determined from experimental measurements of roughness heights as a function of the calculated freezing fraction at the stagnation point. It was not reverse-engineered in order to match ice shape predictions". The relation is:

$$\frac{k_s}{c} = \frac{1}{1000}\frac{1}{2}\sqrt{0.15 + \frac{0.3}{f_0}} \tag{9}$$


where $f_0$ is the freezing fraction at the stagnation point, equal to 1 for rime ice (all the impinging water freezes upon impact). Thus, according to this equation, $\frac{k_s}{c} = 0.34 \times 10^{-3}$ for the case under analysis. For comparison, Shin's relation provides $\frac{k_s}{c} = 0.58 \times 10^{-3}$ for convective heat transfer and $\frac{k_s}{c} = 1.2 \times 10^{-3}$ for drag prediction.

However, these relations do not include the effect of time on $k_s$. They were specifically developed for the atmospheric
conditions typical of aviation, where icing occurs at high wind speeds for a short time. On the other hand, ice accretion on

wind turbines may last for hours. Since the roughness height increases with time, much higher roughness can be found on wind turbine blades. However, such values are currently unknown.

## 2.2  CFD Simulations Setup

The SU2 code solves RANS equations using an edge-based finite volume discretisation in space on general unstructured grids. The convective and viscous fluxes are then evaluated at the midpoint of an edge. An upwind Flux Difference Splitting (FDS) numerical scheme was chosen to solve the convective fluxes in the incompressible solver. Second-order accuracy of the numerical method was obtained by applying a MUSCL scheme for fluxes reconstruction. The gradients of the variables at each node were reconstructed using the Green-Gauss theorem. During reconstruction, gradients were limited using the slope limiter by Venkatakrishnan (1995) to avoid spurious oscillations of the variables.

Steady-state problems are solved with a pseudo-time step technique, in which the solution is marched in time until the time derivative term vanishes and a steady-state solution is reached. An adaptive CFL method was used for convergence acceleration in pseudo-time. Convergence was reached when the root mean square of the residual in the entire domain was reduced at least by three orders of magnitude for all variables, and the normalised relative difference between two consecutive iterations of lift and drag coefficients, averaged over 100 iterations, was smaller than $10^{-6}$.

Each airfoil was simulated approximately between the positive and negative stall angle with a step of $2°$. The Spalart-Allmaras (SA) turbulence model was chosen. The discretisation error was assessed with the Grid Convergence Index (GCI) method by Roache (Celik et al., 2008) by simulating each angle of attack on three different grids. The refinement factor of the grids was $\sqrt{2}$, i.e., the average area of the elements was doubled each time. Then, flow transition was taken into account by applying the algebraic model of Cakmakcioglu et al. (2018) (SA-BC) with a freestream turbulence intensity of 0.1%. Finally, the aerodynamic coefficients were extrapolated to cover the entire range of angles of attack (Viterna and Janetzke, 1982) and corrected for 3D effects (Du and Selig, 1998; Eggers et al., 2003) using NREL's tool *AirfoilPrep*.

Roughness was included in CFD simulations by applying the Boeing extension for the Spalart-Allmaras turbulence model. The rough-wall model was developed by Spalart (Aupoix and Spalart, 2003) and was recently implemented in SU2 (Ravishankara et al., 2020). It is a modification of SA to account for wall roughness so that the logarithmic law of the wall for rough walls (Eq. 6) is correctly represented. It was applied during and after ice accretion. We must point out that this is a low-Re model, i.e., the flow field is computed numerically down to the viscous sublayer, located at $y^+ < 5$, using a turbulence model. This requires $y^+_{\text{wall}} < 1$ to correctly capture the different boundary layer regions. High-Re models, i.e., models using wall functions, would require $y^+_{\text{wall}}$ belonging to the log-law region, i.e., $y^+_{\text{wall}} \approx 50$. Simulations with rough wall functions on iced airfoils can be found in a recent work by Yassin et al. (2021).

An unstructured-hybrid mesh was used to discretise the domain. It was generated using the code uhMesh (Dussin et al., 2009). The circular domain was made of an O-grid of quadrangular elements around the airfoil surrounded by an unstructured grid of triangular elements. uhMesh generates the boundary layer grid with an advancing-front technique with the possibility of local insertion of triangles, while the outer grid is created by computing a Delaunay triangulation using the Bowyer-Watson algorithm. A first cell height of $10^{-6}c$ (chord) ensured that $y^+_{\text{wall}}$ was lower than 1 on the entire airfoil in every simulation.

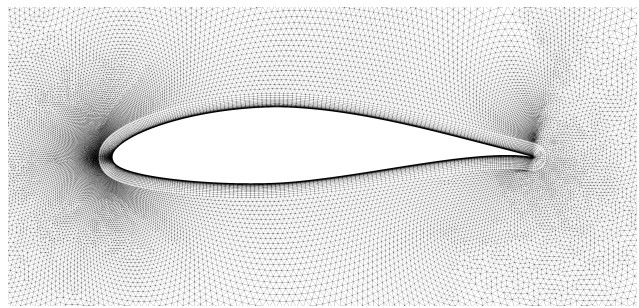

**Figure 2.** Fine grid (NACA 64$_3$-618 airfoil).

A farfield distance of $240c$ ensured the independence of the solution on this parameter. On the finest grid, the characteristic length applied were: $8c$ at the farfield, $0.3c/1000$ near the leading edge, $c/1000$ near the trailing edge, and $c/100$ elsewhere. A close-up view of the fine grid of the NACA 64$_3$-618 airfoil is shown in Fig. 2.

During the icing event, we assumed that the presence of roughness due to ice in the stagnation point region caused the transition to a fully-turbulent flow. Thus, flow transition was neglected from the beginning of the icing event. This modelling hypothesis was necessary since no roughness-induced transition model is currently available in SU2. The fully-turbulent hypothesis may affect two results, i.e., the ice accretion simulations and the aerodynamic coefficients of the iced airfoils.

Regarding flow transition on the iced airfoil, it is interesting to analyse the results of the rough flat plate experiment by Feindt (1957). The experiment is commonly used to verify new roughness-induced transition models with transport equations (Dassler et al., 2010; Langel et al., 2014; Min and Yee, 2021) and their boundary conditions (Son and Kim, 2022). First, let us introduce Feindt's $k_s$ Reynolds number $Re_{k_s} = \frac{\rho U_\infty k_s}{\mu}$. From the experiments, the critical $k_s$ Reynolds number after which the roughness affects transition is $Re_{k_s,cr} = 130$. Moreover, as $Re_{k_s}$ increases, the transition point moves upstream and the width of the transition region decreases. For $Re_{k_s} \gtrsim 300$, the transition point is located at $Re_{x_t} = \frac{\rho U_\infty x_t}{\mu} < 0.1$. On the wind turbine under analysis, considering the outer half of the blade, a rotating velocity of 11 rpm, a minimum $k_s$ of $0.3 \cdot 10^{-3}$ and a maximum $k_s$ of $3 \cdot 10^{-3}$, $Re_{k_s}$ varies from a minimum value of 750 at mid-span to a maximum value of 15000 at the tip. Thus, the fully-turbulent approximation is acceptable for the computation of aerodynamic coefficients.

Regarding ice accretion simulations, Min and Yee (2021) have recently included the effect of roughness-induced transition in ice accretion simulations and compared the results with fully-turbulent rough simulations. The transition model improved the accuracy in glaze ice simulations, while the rime case was unaffected. Indeed, water droplets freeze upon impact with rime ice and the numerical solution mainly depends on the collection efficiency. Thus, the fully-turbulent hypothesis is also appropriate to study a rime ice accretion.

### 2.3 The Ice Accretion Problem

The problem of ice accretion is clearly unsteady. As ice grows on the surface, the shape of the airfoil changes, modifying the flow field, the droplet trajectories, and the ice shape as well. During this process, two time scales can be identified. One is

related to the growth of ice, and the other is related to the modification of the flow field due to ice growth. The former is, in general, much larger than the latter. For this reason, a quasi-steady, multi-step approach must be adopted. The total accretion time is divided into smaller intervals. In each sub-interval, the flow field and droplets trajectories are kept constant, and an ice accretion step is performed. The interaction between the gas and the liquid (droplet) phase can be taken into account by using an Eulerian two-fluid model (Re and Abgrall, 2020; Sotomayor-Zakharov and Bansmer, 2021). Otherwise, a one-way coupling approach can be used by computing the particle trajectories with a Lagrangian approach on the previously computed flow field. After the small ice growth, the geometry is updated and the loop is repeated until the final time is reached. For the geometry update, besides simple re-meshing, conservative mesh adaptation techniques (Cirrottola et al., 2021; Colombo and Re, 2022; Donizetti et al., 2023) or mesh-less immersed boundary methods (Lavoie et al., 2022) can be used to avoid failures in grid generation.

Each of these tasks was performed by different software. Once more, SU2 was used for the computation of the flow field. The Lagrangian particle tracking PoliDrop was used to compute the trajectories of the water droplets and the resulting collection efficiency $\beta$ on the airfoil surface. The ice accretion engine PoliMIce computed the local ice thickness by solving a simplified Stefan problem. Finally, uhMesh was used to generate the grid of the new geometry. No smoothing was applied to the geometries unless grid generation failed. Roughness was applied where ice was predicted using Wright's relation (Eq. 9).

In PoliDrop, we used an iterative method to compute the collection efficiency up to arbitrary precision. The seeding region was updated at each iteration by adding new particles where needed. A uniform seeding front was initialised as a linear grid with equally spaced elements. At the first iteration, the parcels not hitting the airfoil were identified and removed so that the seeding front was reduced in size. The first two parcels flying just above and below the object were not removed so that the impingement limits were refined as well. Then, at each iteration, elements were incrementally split, evolving the current cloud front and computing the collection efficiency $\beta$ on the target surface. The simulation stopped when the difference in the $L_2$ norm between two consecutive iterations of computations of $\beta$ was below a specified threshold:

$$\|\mathrm{err}\beta\|_2 = \left( \sum_{j=1}^{n} \left[ \left( \beta_j^{[k-1]} - \beta_j^{[k]} \right) \Delta s_j \right]^2 \right)^{\frac{1}{2}} < \mathrm{tol} \tag{10}$$

where the index $k$ identifies the iteration number, $j$ is the cell on the airfoil, and $\Delta s_j$ is the size of cell $j$. At each iteration, the number of parcels doubled, $\|\mathrm{err}\beta\|_2$ halved, and the time to complete the iteration doubled as well. To reduce $\|\mathrm{err}\beta\|_2$ by one order of magnitude, approx. 3 to 4 iterations were required, and the computational time increased by a factor of 8 or 16, respectively.

It is clear that the choice of the number of time steps in a multi-step ice accretion simulation and the accuracy of the Lagrangian particle tracking are crucial to obtain an accurate solution efficiently. A proper combination of these parameters is required. These were chosen by comparing the numerical solution with three experimental test cases by Han et al. (2012) at the Adverse Environment Rotor Test Stand (AERTS) of the Pennsylvania State University. The setup was then tested on six additional test cases from the same series of experiments. Results are shown in the Validation section.

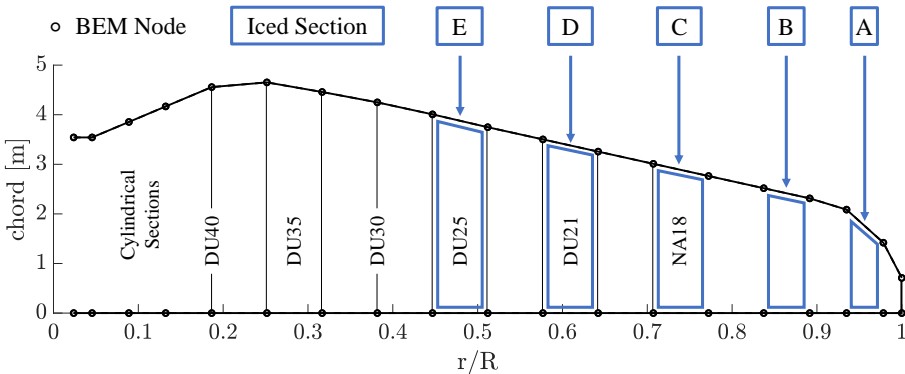

**Figure 3.** Blade discretization and sections chosen for ice accretion.

Once a satisfactory setup for icing simulations was found, the icing event on the full blade was simulated. Icing was monitored on five independent sections, as shown in Fig. 3. Each section was located at the midpoint of two BEM nodes and is representative of the ice that, on average, is accreted on the two nodes. Thus, local boundary conditions on each iced section (i.e., the relative velocity $V_{\mathrm{rel}}$ and the local angle of attack $\alpha$) were computed as the mean value of the two adjacent nodes. Then, the aerodynamic coefficients found on the iced section were applied to the two adjacent nodes. In this way, the entire outer half of the blade was covered. Blade flexibility, wind shear, and tilt and cone angles of the rotor all contributed to generating a periodic output, with a period corresponding to that of a blade revolution, i.e., with a strong 1P component. Since a steady wind was considered, oscillations were limited. In particular, $\Delta\alpha < \pm 0.7$ and $\Delta V_{\mathrm{rel}} < \pm 1\mathrm{ms}^{-1}$. Thus, the mean value of $\alpha$ and $V_{\mathrm{rel}}$ over one period was computed to consider a steady input for ice accretion. In OpenFAST, fully-turbulent aerodynamic coefficients were used as input at this stage to include the effect of early transition due to icing.

The quasi-steady approximation was applied independently to each section, using different time steps according to the local ice accretion rate. The specific time steps used for each section are presented in Sect. 4.1. A matching time of 30 min was chosen to check if it was necessary to update the local boundary conditions due to a change in the equilibrium condition of the wind turbine, similarly to what was done by Zanon et al. (2018). An empirical relation was retrieved using OpenFAST to check the estimated variation in the angle of attack as ice was accreting on the blade sections. However, the difference in $\alpha$ and $V_{\mathrm{rel}}$ was always negligible during ice accretion.

## 2.4 Aerodynamics of the Iced Blade

Due to the high uncertainty in roughness height estimation, two values for $k_s/c$ were considered when computing the aerodynamic coefficients of the iced sections after the icing event. The first one was estimated with Wright's formula (Eq. 9), which generally prescribes $k_s/c = 0.34 \times 10^{-3}$ for rime ice. This roughness height is identified in the text with the letter *W*. Then, this value was increased by one order of magnitude to $3 \times 10^{-3}$. Since this value is close to the one prescribed by Shin's relation corrected for drag prediction, we refer to this case with the letter *S*.

It will be shown in Sect. 3.2 that the impingement limits are slightly under-predicted by PoliMIce and other codes during a steady ice accretion. Moreover, blade vibrations and the highly unsteady incoming wind are likely to increase the wet region of the blade surface during real wind turbine operation. Thus, in numerical studies of wind turbine icing, applying roughness on a greater area than the ice shape only is standard practice. For instance, Homola et al. (2012) used roughness on the entire surface of each airfoil, while Etemaddar et al. (2014) applied roughness to the ice shape and the first 25% of the airfoil chord.

To quantify the effect of this modelling choice on airfoil performance and power losses, besides using two different roughness heights, we also considered two regions to which roughness was applied. In the first case, it was applied to the ice shape only. This case is denoted in the text by adding the subscript *std* (*standard*) to the letter defining the roughness height: $W_{std}$ ($k_s/c = 0.34 \times 10^{-3}$) and $S_{std}$ ($k_s/c = 3 \times 10^{-3}$). The second roughness region included the first one and extended beyond it, covering a length corresponding to 25% of the chord of Section A, i.e. 0.44m, on all sections, on both the upper and the lower surface. On the other sections, from B to E, this corresponded to 18%, 15%, 13%, and 11% of the chord, respectively. This case is denoted in the text by applying the subscript *ext* (*extended*) to the letter defining the roughness height: $W_{ext}$ and $S_{ext}$. The two regions are shown in Fig. 4 on blade Section B. An overview of the four test cases is provided in Table 3. The value of 25% on Section A was picked to match the value chosen by Etemaddar et al.. However, a different modelling choice was made, and this value was kept constant dimensionally (0.44 m) among all sections. By keeping fixed the dimensional width of the *ext* region, its non-dimensional width reduced as the chord of the airfoil increased, being more consistent with the physics of the problem. Indeed, the greater the chord, the greater the pressure gradient generated by the section. Thus, water droplets are deflected away, and the wet area on the section reduces, at least in non-dimensional terms.

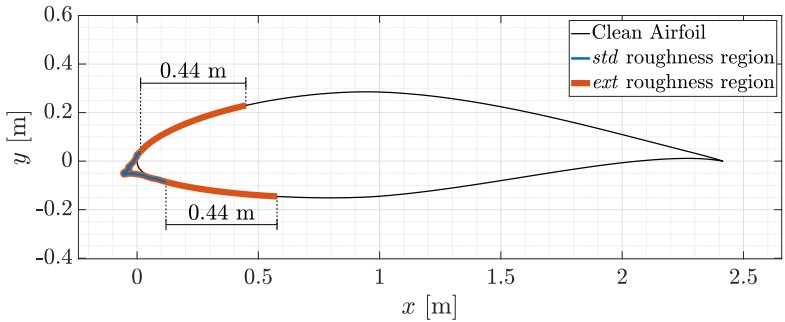

**Figure 4.** Definition of the *std* and of the *ext* roughness regions on an iced airfoil. The *std* region corresponds to the numerically computed ice shape. The *ext* region includes the *std* region and extends for 0.44 m beyond the impingement limits of the ice shape, regardless the chord of the airfoil. On Section B, this corresponds to 18% of the chord.

**Table 3.** Roughness heights and regions tested, and the resulting test matrix.

| Case | Roughness Height $k_s/c$ |
| --- | --- |
| $W_\square$ | $0.34 \times 10^{-3}$ |
| $S_\square$ | $3 \quad \times 10^{-3}$ |

| Case | Roughness Region |
| --- | --- |
| $\square_{std}$ | ice only |
| $\square_{ext}$ | ice+0.44 m |

| | $W_\square$ | $S_\square$ |
| --- | --- | --- |
| $\square_{std}$ | $W_{std}$ | $S_{std}$ |
| $\square_{ext}$ | $W_{ext}$ | $S_{ext}$ |

## 3  Validation of the Numerical Setup

### 3.1  CFD Solver

To validate the setup of the CFD solver, the aerodynamic coefficients of the clean airfoils of the blade were computed and compared with experimental data. The aerodynamic coefficients of DU airfoils were measured by Ruud van Rooij of Delft University of Technology at a Reynolds number of 7 million. NACA 64$_3$-618 coefficients were taken from Abbott et al. (1945) at a Reynolds number of 6 million. All airfoil data are provided in the DOWEC report (Kooijman et al., 2003). The comparison between numerical simulations and experimental results is shown in Fig. $5 - 10$. A correction of $-0.4°$ was applied to experimental data of NA18 airfoil as suggested by Timmer (2009) due to a possible error in the orientation of the model in the wind tunnel. The moment coefficient was computed with respect to $\frac{c}{4}$ and is positive for nose-up.

First, we analyse the results for fully-turbulent flows. All fully-turbulent simulations showed satisfactory grid convergence for almost every aerodynamic coefficient computed. The estimate of the discretisation error was computed with the GCI method (Celik et al., 2008) and was represented through error bars. In the attached flow regime, the lift coefficient was underestimated on all airfoils, while the drag coefficient was overestimated. The error with respect to experimental data increased together with the relative thickness of the airfoils. It was maximum at the root of the blade. The positive stall angle and lift coefficient were over-predicted for $t/c \leq 30\%$. The maximum lift coefficient became under-predicted for $t/c \geq 35\%$, with the error increasing for increasing $t/c$. At negative stall, good predictions were made up to $t/c \leq 21\%$, while minimum lift coefficients were underestimated (in absolute value) for $t/c \geq 30\%$.

When the algebraic BC transition model was included in the system of equations, the aerodynamic coefficients were accurately predicted on all airfoils for attached flows, regardless of their relative thickness. The absolute value of the maximum lift coefficient increased, both at positive and negative stall. This led to more accurate predictions of positive stall for $t/c \geq 35\%$ and of negative stall for $t/c \geq 25\%$.

The Boeing extension for Spalart-Allmaras was then tested to compare the numerical results with the law of the wall for rough surfaces. The relation was presented in Eq. 6. Two cases were considered. In the first case, a roughness height $k_s/c = 0.5 \times 10^{-3}$ was applied on the entire surface of the NA18 airfoil. In the second case, $k_s/c$ was set to $5 \times 10^{-3}$. Each value of $k_s/c$ leads to various $k_s^+$. This occurs since the skin friction varies locally, and so does the viscous length scale $\delta_\nu$. In these simulations, the Reynolds number was 6.6 million and the angle of attack was $0°$. The results are shown in Fig. 11. The velocity profile in wall units $u^+$ are shown as a function of the non-dimensional wall distance $y^+$ at different stations along the airfoil, and compared to the theoretical results obtained with the local $k_s^+$. On the top row, results are for $k_s/c = 0.5 \times 10^{-3}$ on the

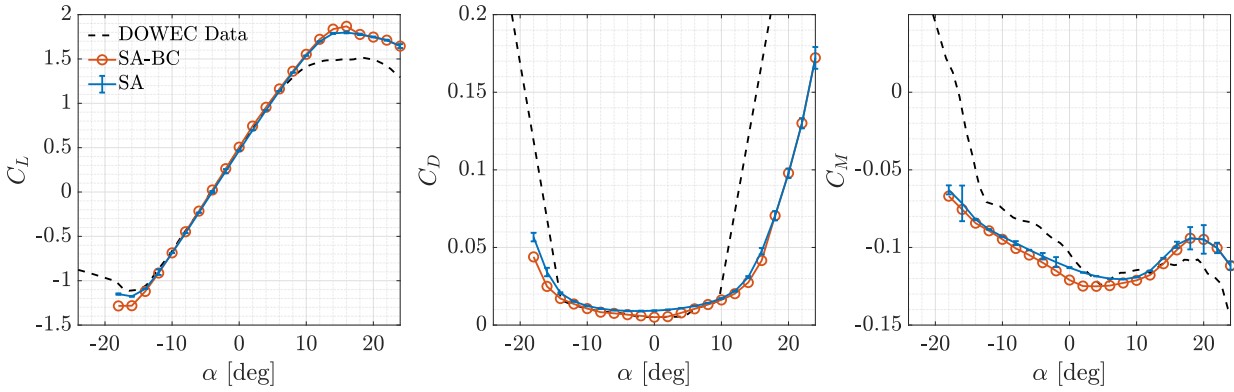

**Figure 5.** Aerodynamic coefficients of NA18 airfoil.

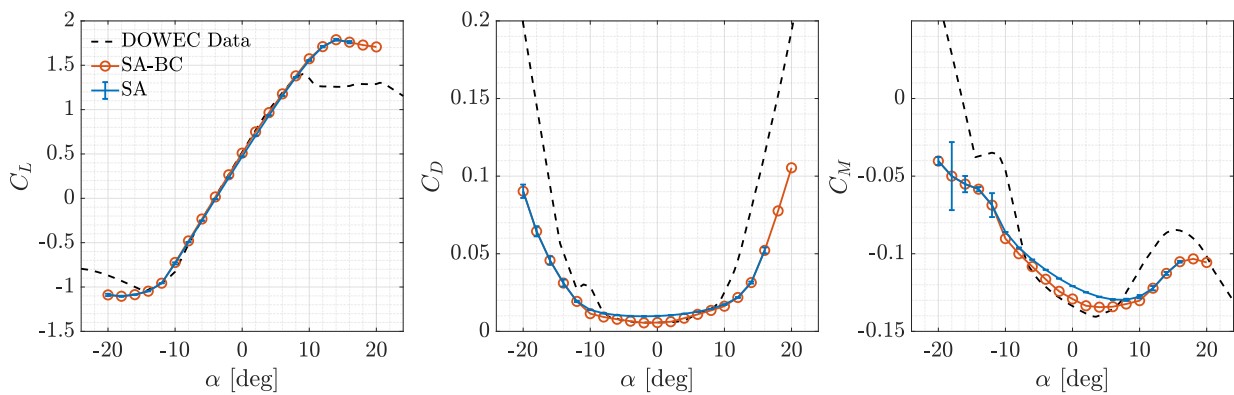

**Figure 6.** Aerodynamic coefficients of DU21 airfoil.

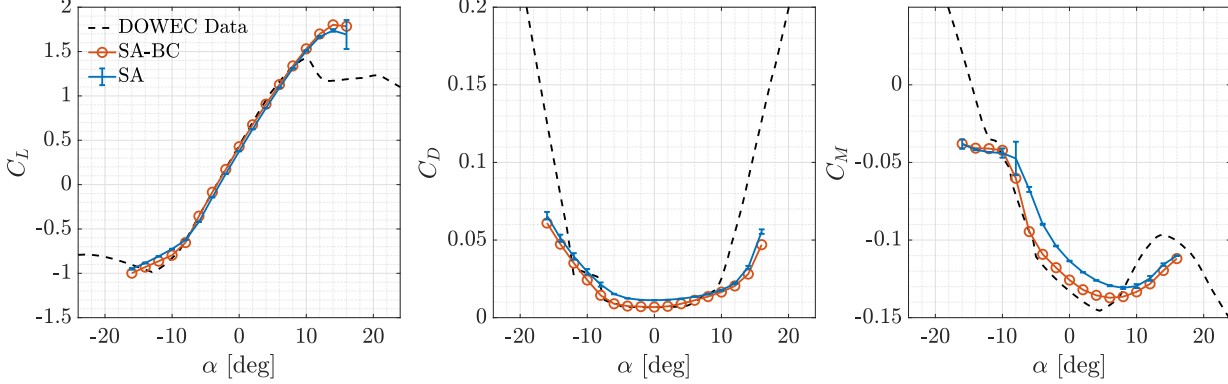

**Figure 7.** Aerodynamic coefficients of DU25 airfoil.

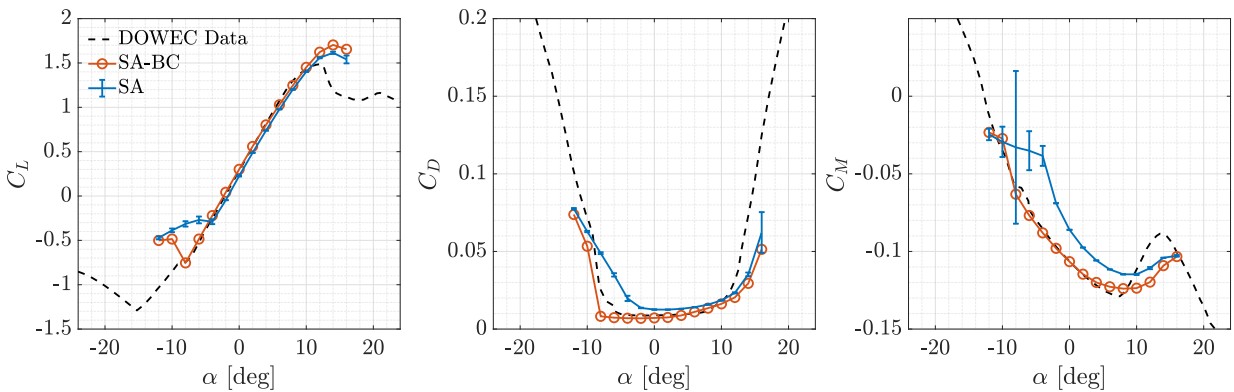

**Figure 8.** Aerodynamic coefficients of DU30 airfoil.

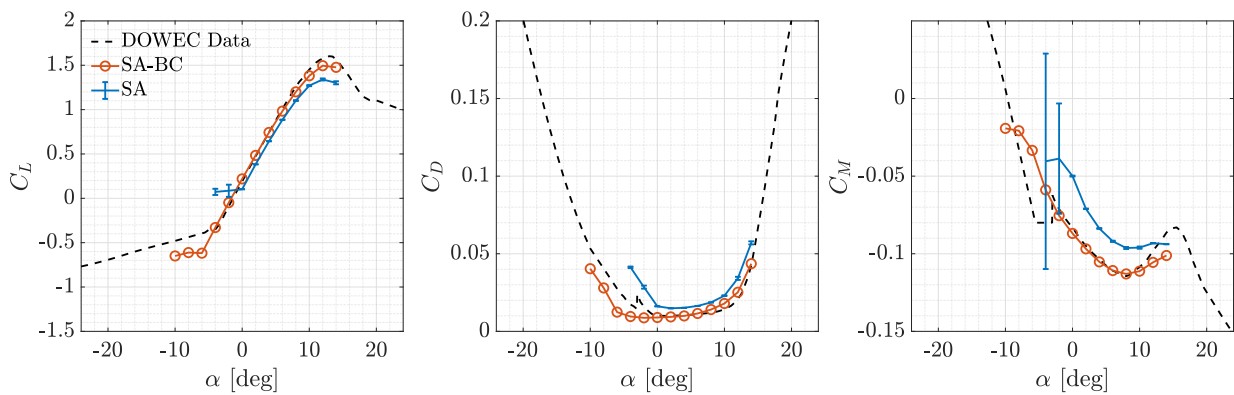

**Figure 9.** Aerodynamic coefficients of DU35 airfoil.

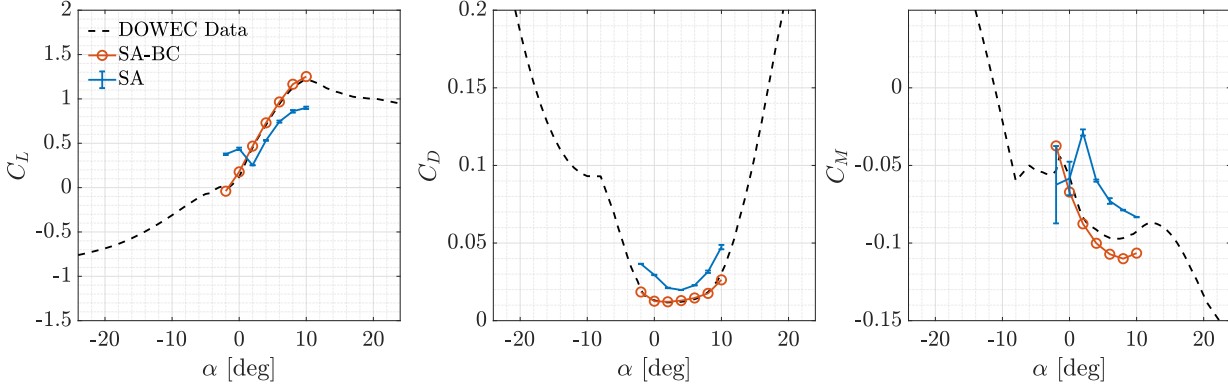

**Figure 10.** Aerodynamic coefficients of DU40 airfoil.

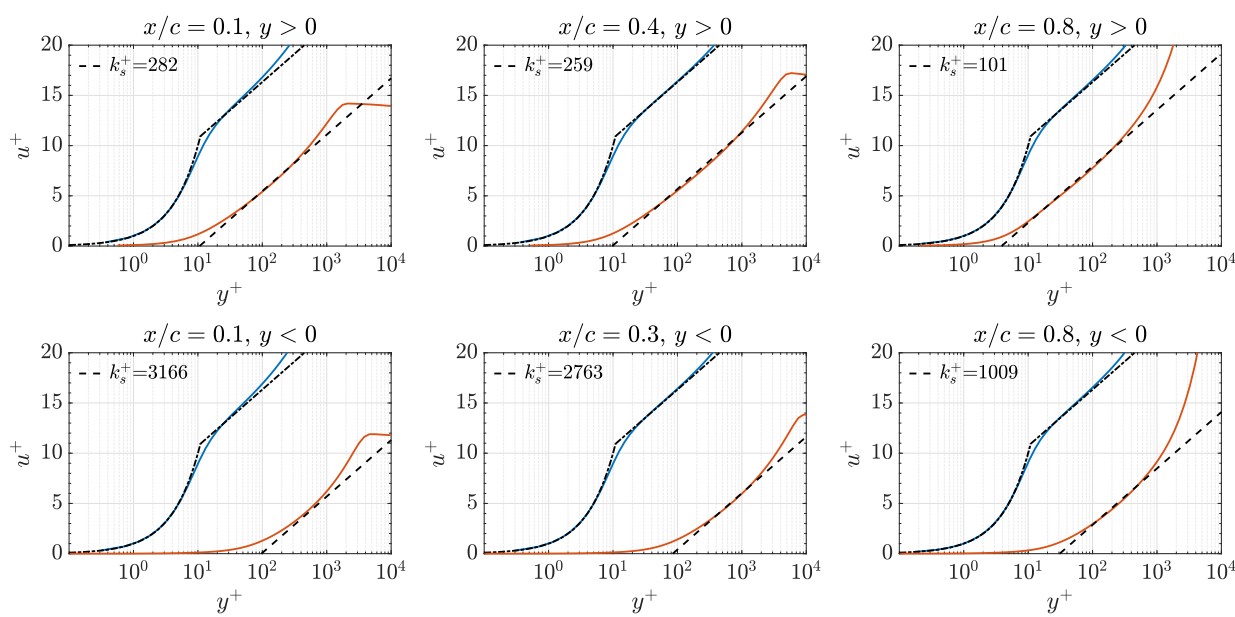

**Figure 11.** Law of the wall using Boeing extension for Spalart-Allmaras turbulence model. NA18 airfoil, $\alpha = 0°$, $Re = 6.6$ million. Top row: $k_s/c = 0.0005$. Bottom row: $k_s/c = 0.005$.

suction side of the airfoil. On the bottom row, $k_s/c = 5 \times 10^{-3}$ and results are extracted from the pressure side. The numerical solution for the smooth airfoil is also shown and compared with the theoretical behaviour (Eq. 8). For the values of $k_s/c$ under analysis, all the resulting $k_s^+$ belonged to the fully-rough regime, typical of ice. The model accurately captured the different shifts in the logarithmic region of the law of the wall.

## 3.2 Ice Accretion Simulations

Two different approaches were tested for ice accretion. These were almost equivalent in overall computational time. In the first one, the collection efficiency $\beta$ of the droplets was finely computed during each time step, setting $\|\text{err}\beta\|_2 < 3 \times 10^{-6}$ (Eq. 10). In the second one, the tolerance was set to $3 \times 10^{-5}$ and the number of icing steps was increased. Numerical results were compared with experimental rime ice accretion on a rotating blade section, consisting of an S809 airfoil with $c = 0.267$ m. Experiments were carried out by Han et al. (2012). AERTS test cases #20, #21, and #22 were chosen to find the best computational setup. Then, the chosen setup was tested on six additional cases, i.e., AERTS #4 and #15-19. Test cases #20-22 were chosen as the primary benchmark since they are the longest available and test conditions are the most similar to those of the icing event under analysis. The three test cases only differ in the duration of the icing event, which was 30 min, 60 min, and 90 min, respectively. Test conditions are reported in Table 4.

**Table 4.** Test conditions of AERTS test cases.

| Case # | MVD [µm] | LWC [gm$^{-3}$] | $T$ [°C] | $V_{\text{rel}}$ [ms$^{-1}$] | AoA [deg] | Time [min] |
|--------|----------|------------------|----------|------------------------------|-----------|------------|
| 4 | 20.0 | 0.08 | -7.0 | 50.0 | 2.0 | 30 |
| 15 | 20.0 | 0.08 | -9.0 | 50.0 | 4.0 | 30 |
| 16 | 20.0 | 0.08 | -4.5 | 50.0 | 2.0 | 30 |
| 17 | 20.0 | 0.08 | -7.0 | 50.0 | 4.0 | 30 |
| 18 | 20.0 | 0.05 | -10.0 | 50.0 | 4.0 | 30 |
| 19 | 20.0 | 0.05 | -9.0 | 50.0 | 8.0 | 30 |
| 20 | 20.0 | 0.05 | -9.0 | 50.0 | 4.0 | 30 |
| 21 | 20.0 | 0.05 | -9.0 | 50.0 | 4.0 | 60 |
| 22 | 20.0 | 0.05 | -9.0 | 50.0 | 4.0 | 90 |

A time step of 15 min was used when $\|\text{err}\beta\|_2 = 3 \cdot 10^{-6}$, while 3 min was chosen when $\|\text{err}\beta\|_2 = 3 \cdot 10^{-5}$ to try to match the computational time. Results are reported in Fig. 12. In both cases, the ice impingement limit on the lower surface was underestimated. This is a common issue in numerical ice accretion simulations. A real cloud is made by a distribution of droplet diameters and the MVD is just an indicator of the median of this distribution. Parcels with higher diameters have a higher mass, and the pressure gradient deflects their trajectory less. Thus, a wider portion of the airfoil gets wet. This phenomenon may be overcome in numerical simulations with a multi-bin approach, i.e., by performing the weighted average of the collection efficiency computed with different droplet diameters from the diameters distribution (Sirianni et al., 2022). The effect of the uncertainty in other operating conditions was analysed in a recent work by Gori et al. (2022) On the other hand, by using a finer time discretization, a more accurate solution at the leading edge was obtained. A higher number of ice layers led to a better representation of the physics of the problem while limiting the propagation of errors from one step to the other. This permitted the reduction of the accuracy in the computation of $\beta$ without losing accuracy in the computation of the solution. Moreover, a noticeable reduction of the elapsed real time for the entire 90 min simulation was found (approx. 13%). For these reasons, the approach consisting of a high number of time steps with lower accuracy on $\beta$ was chosen.

The setup with $\|\text{err}\beta\|_2 = 3 \cdot 10^{-5}$ was tested on the remaining AERTS test cases. The time step was adjusted according to the LWC. For test cases #18 and #19, $\Delta t = 180$ s was used once more. For test cases #4 and #15-17, $\Delta t = 112.5$ s was chosen to match the accumulation parameter LWC $\cdot \Delta t$ of all the previous simulations. Results are shown in Fig. 13. The results obtained with the selected setup show an almost perfect match with LEWICE simulations. Good agreement is found with the experiments in terms of ice thickness and impingement limits on the upper surface. On the lower surface, impingement limits are at times underestimated.

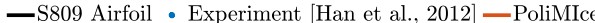

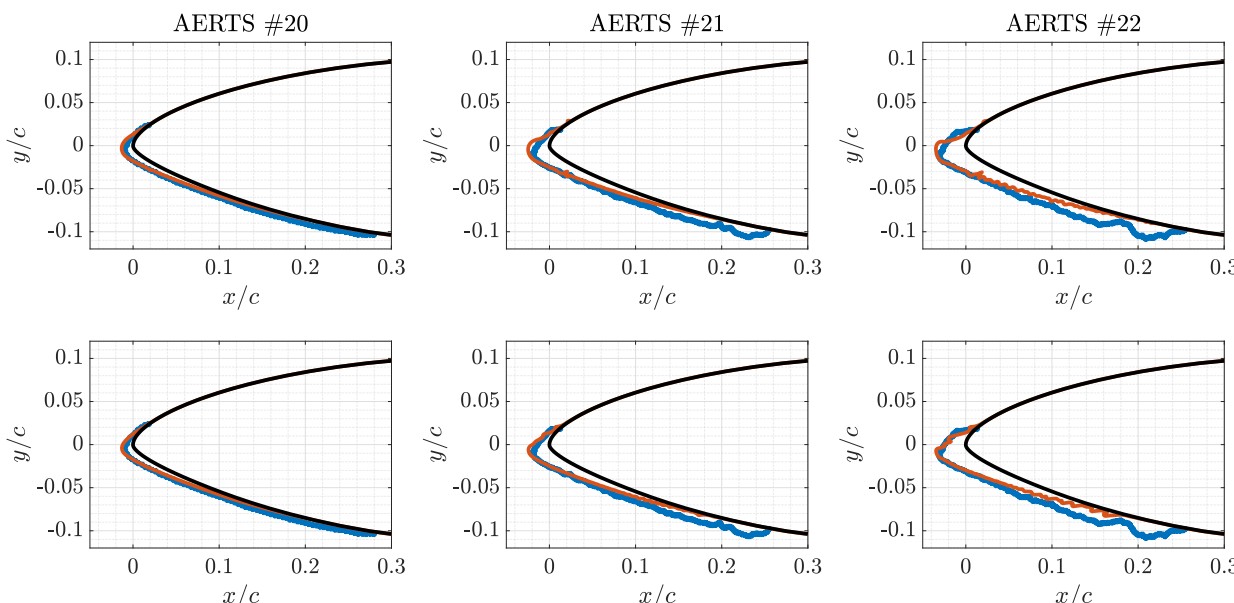

**Figure 12.** PoliMIce simulations of three AERTS test cases. Each column represents the same test case. The three test cases have the same atmospheric conditions and only differ in total ice accretion time (left: 30 min; centre: 60 min; right: 90 min). In each row the same numerical setup is used for the multi-step ice accretion (top: $\Delta t = 15$ min, $\|\text{err}\beta\|_2 < 3 \cdot 10^{-6}$; bottom: $\Delta t = 3$ min, $\|\text{err}\beta\|_2 < 3 \cdot 10^{-5}$).

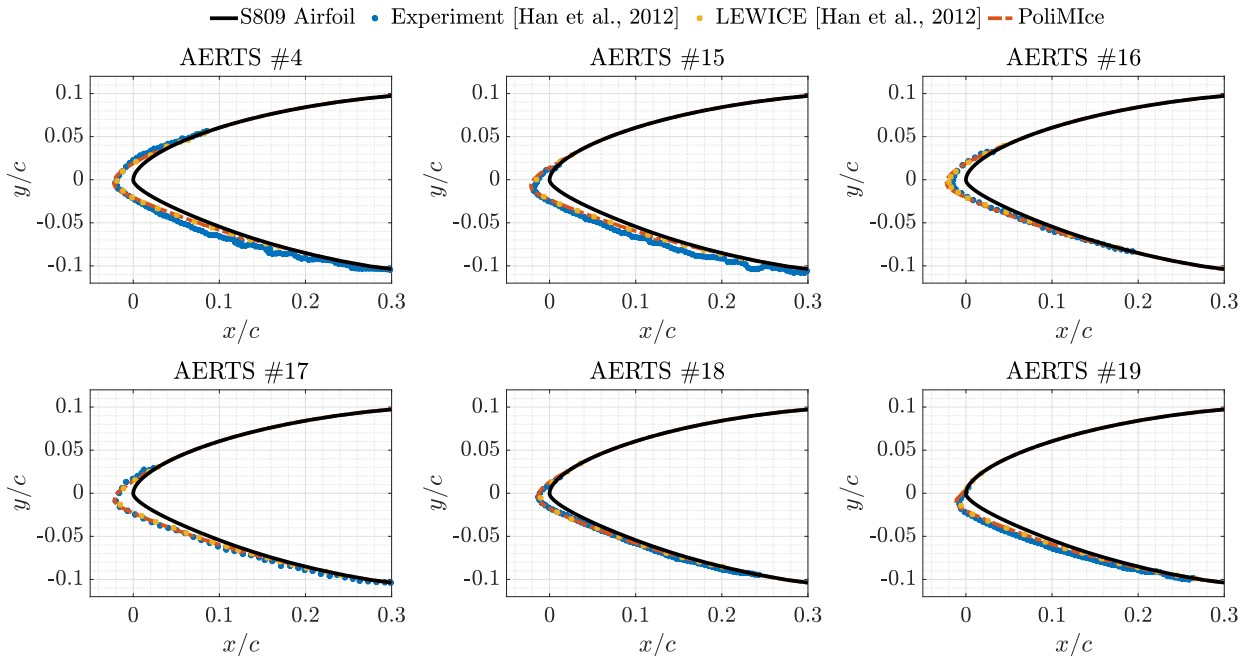

**Figure 13.** Comparison between PoliMIce simulations, LEWICE simulations and experiments of AERTS test cases. $\|\text{err}\beta\|_2 < 3 \cdot 10^{-5}$.

**Table 5.** Local boundary conditions on the five sections under analysis.

| Section | Airfoil ID | $r/R$ [−] | chord [m] | $V_{rel}$ [ms$^{-1}$] | AoA [deg] | $\Delta t$ [min] |
|---------|-----------|-----------|-----------|-----------------------|-----------|------------------|
| A | NA18 | 0.93 | 1.753 | 72.75 | 3.80 | 1 |
| B | NA18 | 0.84 | 2.416 | 65.95 | 3.96 | 2 |
| C | NA18 | 0.72 | 2.887 | 56.71 | 3.85 | 3 |
| D | DU21 | 0.59 | 3.379 | 47.08 | 3.68 | 6 |
| E | DU25 | 0.46 | 3.878 | 37.51 | 4.26 | 15 |

## 4 Results and Discussion

### 4.1 Blade Icing

The local boundary conditions computed at the beginning of the icing event are reported in Table 5, together with the time step chosen for each section. There was no need to update the boundary conditions during ice accretion. For instance, the angle of attack of Section B increased by 0.35° after the icing event due to the degradation of the aerodynamic performances of the wind turbine. However, this may not hold if greater roughness height and extension values were considered during ice accretion. The computed ice shapes are shown in Fig. 14 in non-dimensional form, while a detailed view of the multi-step process on Section B is shown in Fig. 15. The ice shapes on Sections A, B, and C (i.e., NA18 Sections) were very similar. Their main difference was the length of the horn, which decreased towards the root of the blade. Some small secondary protrusions were formed on the main ice shape. These are due to some small oscillations of the collection efficiency, which eventually got amplified step after step because the geometry was not smoothed unless strictly required by the grid generator. Section E was almost unaltered. On this section, $0.42\ \text{kgm}^{-1}$ of ice was found. The ice mass accreted on the blade increased almost linearly, up to $3.35\ \text{kgm}^{-1}$ on Section A. The total accreted mass was estimated to be lower than 100 kg, i.e. less than 0.5% of the total mass of the blade. Thus, it was chosen to neglect the additional mass during the aeroelastic simulations, although its distribution may have altered the modal response of the wind turbine.

### 4.2 Iced Blade Aerodynamics

The aerodynamic coefficients of the iced sections were then computed in the four cases defined in Sect. 2.4, i.e., $W_{std}$, $S_{std}$, $W_{ext}$, and $S_{ext}$ (see Table 3 and Fig. 4). Before the computation, the size of the concave regions around the secondary protrusions in Sections A and B was reduced to improve grid quality while maintaining the same overall ice shape. The result of this process is shown in Fig. 15 for Section B by superimposing the computational grid used for the aerodynamic coefficients onto the ice shape coming from the ice accretion simulations. A red box identifies the most critical region for grid generation, which is shown in detail in Fig. 16, together with the orthogonality of the grid. uhMesh guaranteed good quality meshing by generating a hybrid boundary layer grid, introducing triangles in highly concave and convex regions.

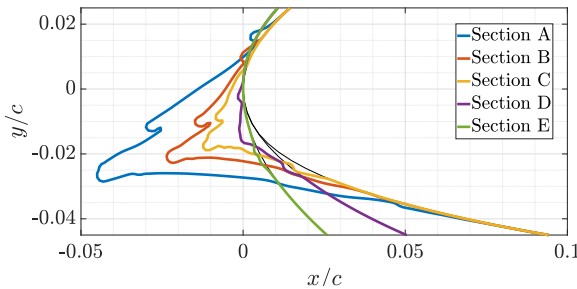

**Figure 14.** Non-dimensional comparison of the ice shapes on sections A-E.

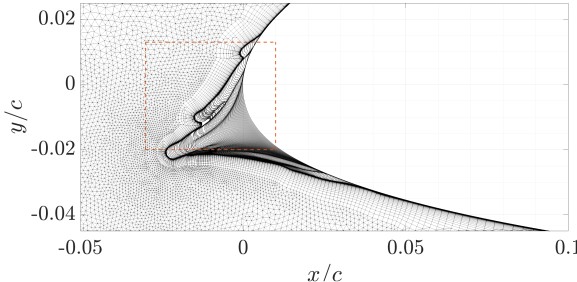

**Figure 15.** Multi-step ice accretion on Section B. The grid for the computation of the aerodynamic coefficients is superimposed onto the computed ice shape, highlighting the removal of highly concave regions. The region within the red box is enlarged in Fig. 16.

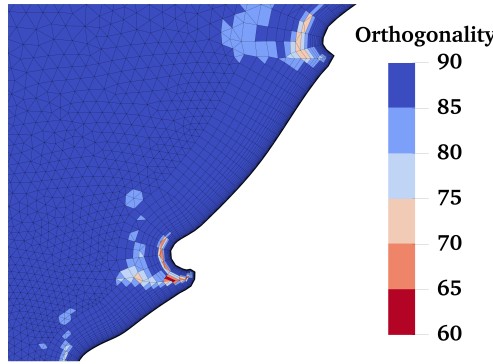

**Figure 16.** Detail of the computational grid on the final ice shape of Section B, showing the orthogonality of the grid. The region corresponds to the red box shown in Fig. 15

Results of the CFD simulations are shown in Fig. $17 - 21$, considering the effect of roughness height and extension. Quantitative comparison is provided in Table 6, where the percentage variation in the aerodynamic coefficients with respect to the clean case is reported for each section and each roughness at $\alpha = 4°$. The aerodynamic coefficients were non-dimensionalised with respect to the clean airfoil chord. The moment coefficient was computed with respect to the same point of the clean airfoil ($c_{\text{clean}}/4$). In all cases, the presence of ice caused a degradation of the aerodynamic performances due to both the ice shape and

roughness. As expected, stall was anticipated, the slope of the lift coefficient decreased, the drag coefficient increased, and the moment coefficient changed significantly. The greatest difference was found when a higher roughness was applied to a wider portion of the airfoils ($S_{\text{ext}}$). The case with smaller roughness on the same region followed ($W_{\text{ext}}$). When roughness was applied where ice was predicted ($W_{\text{std}}$ and $S_{\text{std}}$), the results were similar. However, the behaviour of each section was different. We may analyse the results by distinguishing between the effects of ice shapes and roughness.

We start considering the $W_{\text{std}}$ and $S_{\text{std}}$ cases. Given the decreasingly big ice horns and the small difference between the two cases on Sections A, B, and C, we can conclude that the ice shape was mainly responsible for the aerodynamic penalty on NA18 sections. This can be clearly seen in Fig. 22, where the results for the ice shape without roughness were included. In the figure, the lift-to-drag ratio of Section B is represented as a function of the angle of attack. The efficiency of the section considering a smooth ice shape was almost coincident with $W_{\text{std}}$ case, and only a slight decrease was found with a higher roughness on

the ice shape ($S_{\text{std}}$). For completeness, the results of the clean, smooth, fully-turbulent airfoil were included to qualitatively highlight the effect of icing at the beginning of ice accretion, when the ice shape is negligible, and transition occurs earlier and earlier due to increased roughness. On the other hand, on DU sections the ice shape was small, and so was the region where roughness was applied. For attached flows, results were almost identical to those of the respective clean airfoils with a fully-turbulent flow. Moreover, in this range of AoAs, the performance degradation on the thicker Section E was slightly higher

compared to the thinner Section D. This is coherent with the results of the fully-turbulent clean airfoils. For these reasons, we can conclude that the difference between the clean and the two iced cases in this flow regime was simply due to the early transition for the presence of roughness, which was modelled with a fully-turbulent flow, rather than roughness height or the ice shape. In reality, roughness height affects transition, but it was shown at the end of Sect. 2.2 that the assumption of a fully-turbulent flow is reasonable for such a long ice accretion. On these same sections, positive stall was almost unaffected

compared to the fully-turbulent solution. Negative stall occurred earlier, in particular on Section D. This effect was related to the small, downward-pointing ice shape. Given these results, we can conclude that the effect of ice shape becomes predominant over roughness as the horn grows in size, in accordance with previous studies (Battisti, 2015).

We now consider the two cases of extended roughness: $W_{\text{ext}}$ and $S_{\text{ext}}$. For these two cases, the results were different from each other and were also different from the *std* cases in almost every simulation. We highlight once more that the region of

extended roughness was equal in size ($0.44$m) on all sections, and so it increased in terms of non-dimensional airfoil length from root to tip. By looking at NA18 sections, at the lowest angles of attack, the aerodynamic coefficients coincided in all cases. Thus, at these AoAs, the aerodynamic penalties were still produced by the ice shapes. Negative stall occurred at $\alpha = -8°$ when the separated regions generated from the leading and the trailing edge merged. As the angle of attack increased, the effect of roughness extension became more and more important, together with the value of $k_s$. The slope of the $C_L(\alpha)$ curves

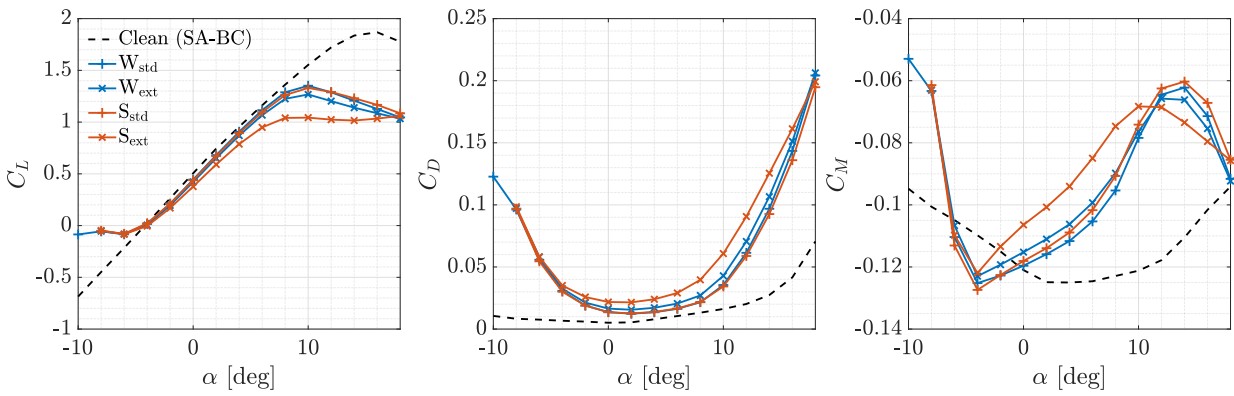

**Figure 17.** Aerodynamic coefficients of Section A.

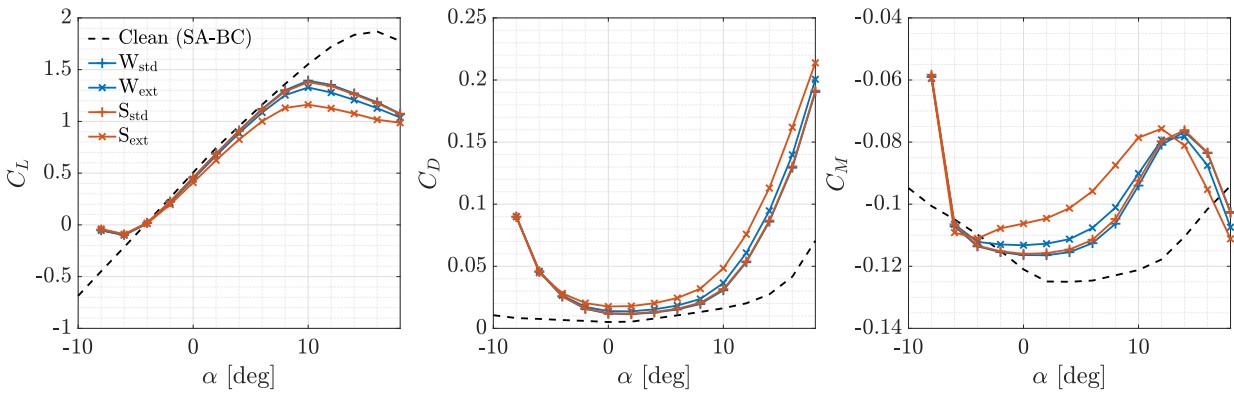

**Figure 18.** Aerodynamic coefficients of Section B.

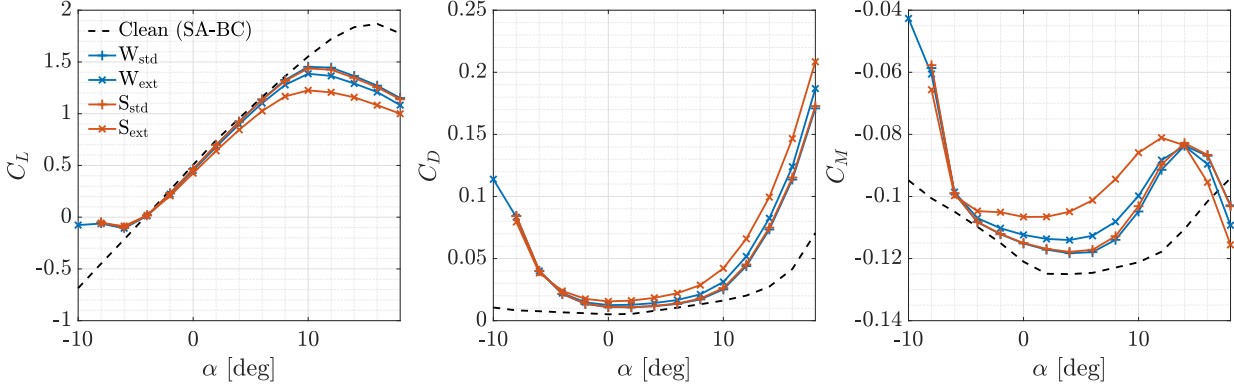

**Figure 19.** Aerodynamic coefficients of Section C.

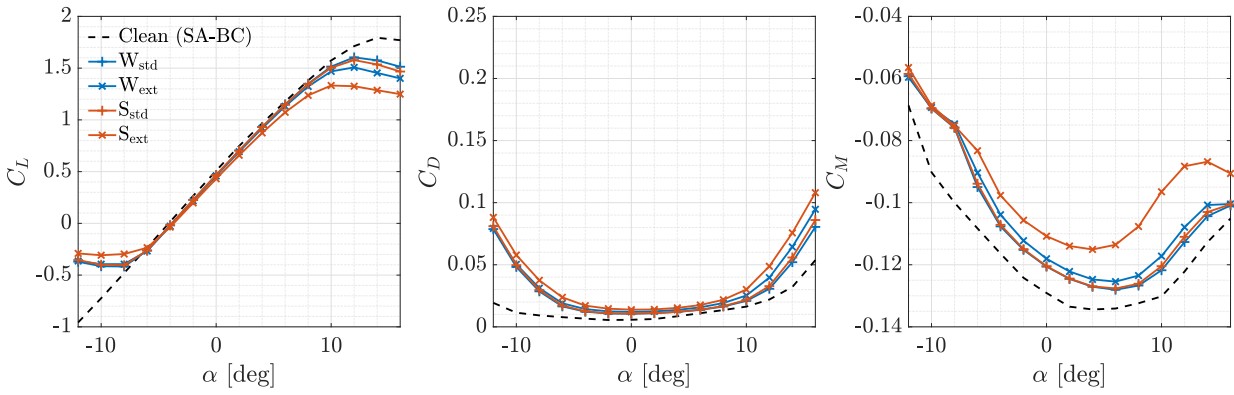

**Figure 20.** Aerodynamic coefficients of Section D.

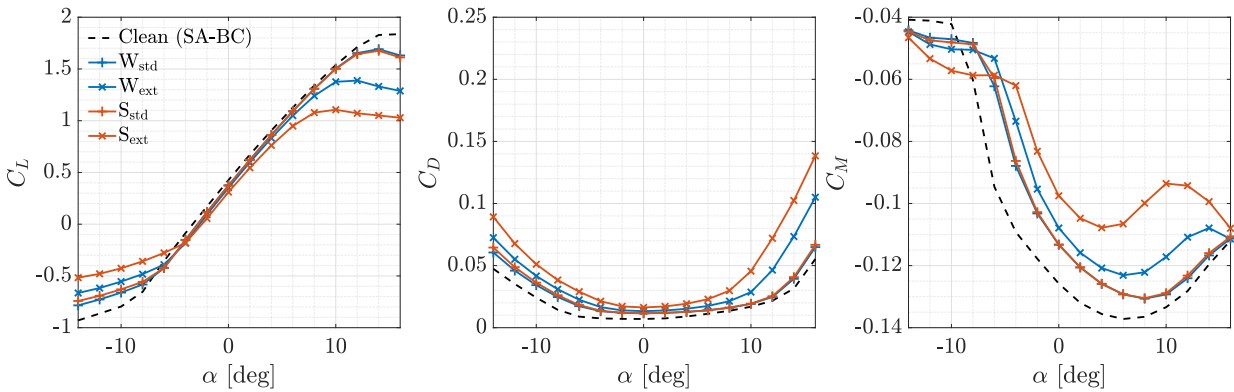

**Figure 21.** Aerodynamic coefficients of Section E.

**Table 6.** Aerodynamic penalties on Sections A-E at $\alpha = 4°$.

| Section | $\Delta C_L$ | | | | $\Delta C_D$ | | | | $\Delta C_M$ | | | |
|---------|--------------|--------------|--------------|--------------|--------------|--------------|--------------|--------------|--------------|--------------|--------------|--------------|
| | $W_{std}$ | $S_{std}$ | $W_{ext}$ | $S_{ext}$ | $W_{std}$ | $S_{std}$ | $W_{ext}$ | $S_{ext}$ | $W_{std}$ | $S_{std}$ | $W_{ext}$ | $S_{ext}$ |
| A | -5.1% | -6.5% | -8.9% | -17.6% | +72% | +75% | +117% | +204% | -10.7% | -12.9% | -15.0% | -24.8% |
| B | -4.4% | -4.9% | -7.1% | -13.6% | +61% | +67% | +94% | +156% | -7.6% | -8.3% | -11.0% | -19.0% |
| C | -3.2% | -3.5% | -6.0% | -11.6% | +48% | +50% | +79% | +132% | -5.4% | -5.6% | -8.7% | -16.1% |
| D | -3.5% | -3.6% | -4.9% | -9.5% | +41% | +42% | +60% | +80% | -5.4% | -5.5% | -7.2% | -14.4% |
| E | -4.8% | -4.8% | -8.1% | -15.9% | +41% | +41% | +69% | +114% | -7.2% | -7.2% | -10.9% | -20.5% |

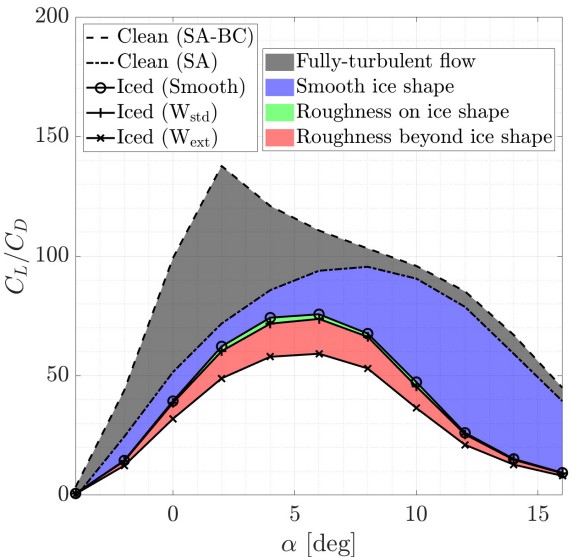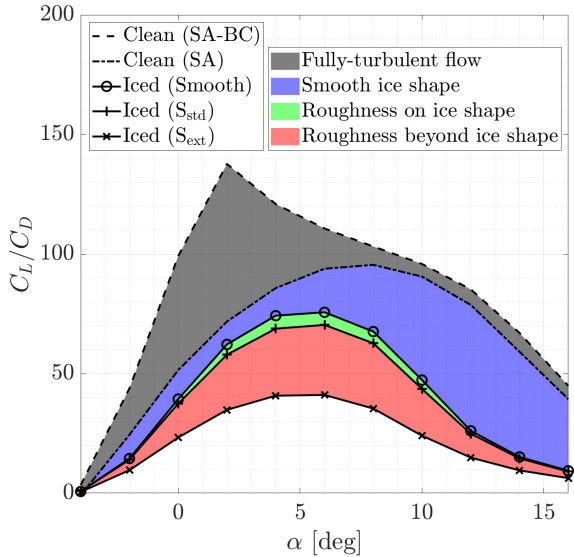

**Figure 22.** Lift-to-drag ratio of Section B considering different cases. Lines represent airfoil efficiency in various cases. Shaded areas represent different contributions to loss in efficiency, providing a qualitative superposition of effects. Left: $k_s/c = 0.34 \cdot 10^{-3}$. Right: $k_s/c = 3 \cdot 10^{-3}$.

decreased while drag and moment coefficients increased. This effect is peculiar since roughness should have little effect on the aerodynamic coefficients when ice horns are well developed. The extended roughness region caused a high increase in skin friction in a geometrically smooth region of the section, increasing the viscous drag. Moreover, the flow expanded less on the suction side of the sections and was compressed less on the pressure side, causing a noticeable reduction in lift. The differences were much higher when roughness was increased by one order of magnitude (i.e., for case S). The difference between the *std* and the *ext* cases increased towards the tip of the blade since roughness was applied on a wider portion of the airfoil. On DU sections, however, the opposite occurred. For both the roughness heights tested, Section E was more sensitive to roughness than Section D in both attached flow and stall conditions. This occurred despite roughness covering a slightly shorter portion of the innermost section. Previously, it was shown that the ice shape only affects the negative stall of Section D. In the other flow regimes of Section D, and on all flow regimes of Section E, we may think to have a fully-turbulent airfoil, i.e., an airfoil where transition is fixed at the stagnation point. For a transition-fixed flow, Somers (2005) found that the detrimental effect of leading edge roughness increases with the relative thickness of the airfoil, as occurs in this case.

### 4.3 Effect of Icing on Power Production

In the previous section, it was shown that the differences between $W_{std}$ and $S_{std}$ cases are negligible. Moreover, only small differences are found with the $W_{ext}$ case. Thus, in this section, only the lower-roughness, tighter-impingement case ($W_{std}$) and the higher-roughness, wider-impingement case ($S_{ext}$) are compared with the clean case (SA-BC).

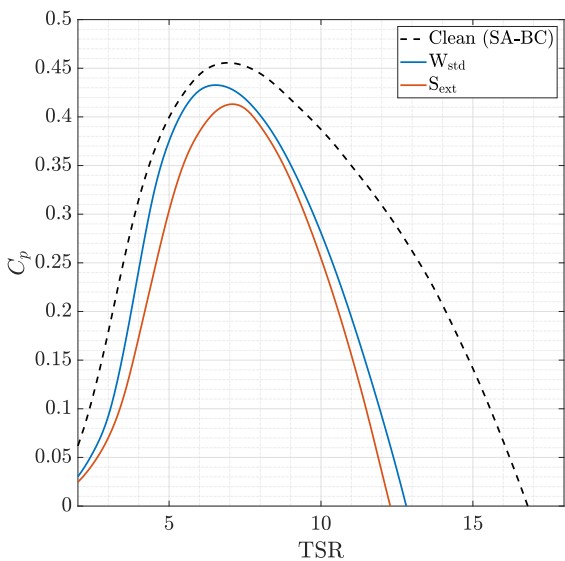

**Figure 23.** $C_P - TSR$ curves with pitch angle $\beta = 0°$.

The $C_P$-TSR curves (Eq. 2) were computed for a pitch angle of 0° using the aerodynamic module of OpenFAST, AeroDyn. Results are shown in Fig. 23. In the iced cases, the $C_P$ values were lower for any TSR. As expected, the lowest values were found in the $S_{\text{ext}}$ case. The highest decrease in $C_P$ occurred at TSR $> 7$. These values are used at low wind speeds when the wind turbine operates in Region 1.5. In particular, from the cut-in wind speed up to 8 m/s, the TSR decreases from 15.3 to
its optimum value of $\sim 7$. The power coefficient became negative for TSR values between 12 and 13. It is worth noticing how the TSR corresponding to the maximum $C_P$ changed from case to case. It was approximately 7 for the clean case, while it decreased to 6.5 for $W_{\text{std}}$ and it increased to 7.1 for $S_{\text{ext}}$.

Then, the power curves were computed with a steady inflow. They are shown in Fig. 24. Power losses are shown in Fig. 25 both as absolute and normalised differences with respect to the clean case. With ice, power production started at 4 ms$^{-1}$. The
normalised power loss was maximum at cut-in wind speed and diminished as the TSR decreased from the start-up value of approximately 15 to a constant value in Region 2. In this region, the power loss is approximately 6% for $W_{\text{std}}$ and 9% for $S_{\text{ext}}$. The TSR value obtained through the generator torque controller in Region 2 was approximately 7.4 in the clean case and 7.2 in the iced cases. This means that an almost optimum TSR was used for $S_{\text{ext}}$, while a sub-optimum one was used for both $W_{\text{std}}$ and the clean case. By regulating the generator torque, their power output may increase.

Next, the power curves were computed with the turbulent inflow prescribed by the IEC in DLC 1.1. They are shown in Fig. 26, while power losses are shown in Fig. 27. The first clear effect of the increased turbulence intensity is the inflexion of the power curve close to the rated speed. On the other hand, the non-constant, non-uniform wind speed made the wind turbine produce slightly more power at low wind speeds. Regarding power losses, with a mean wind speed of 3 ms$^{-1}$ a slightly higher power was produced with ice with respect to the clean case. This result may differ if a different random seed was used to

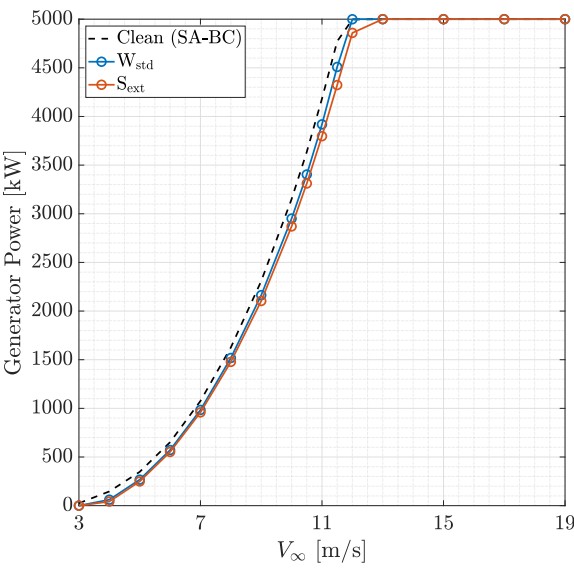

**Figure 24.** Power curve with a steady inflow.

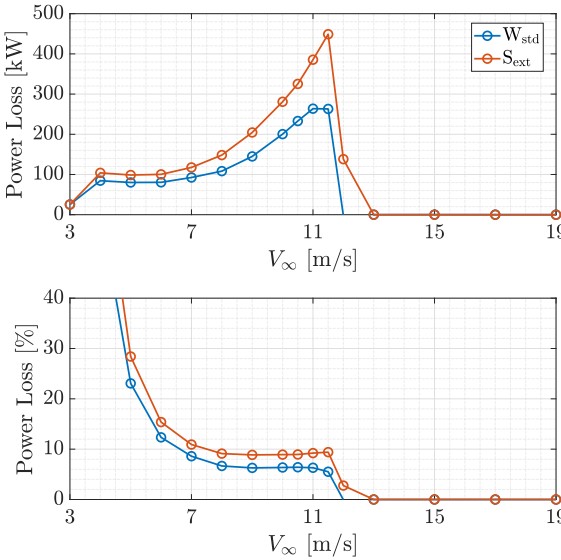

**Figure 25.** Power losses with a steady inflow. Top: absolute difference. Bottom: normalised difference.

**Table 7.** Weibull-averaged power loss computed for the current study and compared with other authors.

| Author | Icing time [h] | $1 - P_W^{iced}/P_W^{clean}$ |
|---|---|---|
| Current study ($W_{std}$) | 3 | 3.44% |
| Current study ($S_{ext}$) | 3 | 5.16% |
| Homola et al. (2012) | 1 | 11.6% |
| Turkia et al. (2013) | 3.33 | 10.7% |
| Etemaddar et al. (2014) | 24 | 14.7% |

generate the realisation of the turbulent wind used as input. At higher wind speeds, the trend was similar to that of a steady inflow. However, due to the variability of wind, there was no clear distinction between the different controller regions, and the results of steady wind were smoothed out. In general, higher power losses were found at any wind speed, except for the nominal rated speed (11 $\mathrm{ms}^{-1}$). Power losses in turbulent wind reduced and became null at approx. 15 $\mathrm{ms}^{-1}$. The actual rated speed remained unchanged after the icing event at 17 $\mathrm{ms}^{-1}$.

As visible from Figures 25 and 27, the effect of roughness on power production depended on wind speed. At low wind speeds, the power loss was similar in both cases under analysis, while differences increased with wind speed. This trend was aligned with the one found by Etemaddar et al. (2014), while it differed from those found by Homola et al. (2012) and Turkia et al. (2013). In order to give a single figure of the difference between the two roughness cases, the Weibull-averaged power $P_W$ was computed and compared with the clean case. Its value was 2618 kW, 2528 kW, and 2483 kW, for the clean, $W_{std}$, and $S_{ext}$

cases, respectively. The average power loss of $W_{std}$ was 3.44%. For the $S_{ext}$ case, it was 5.16%, which is 50% higher than $W_{std}$. This difference is not negligible, even though the ice shapes were well developed and the region of extended roughness was rather limited. The same quantity was computed for the power curves computed by Homola et al. (2012), Turkia et al. (2013), and Etemaddar et al. (2014) and was reported in Table 7. Once more, our results were consistent with those by Etemaddar et al., where the icing event lasted 24h and roughness was applied on 25% of the chord of the blade. On the other hand, Homola

et al. predicted an average power loss of about 10% for an icing event of one-third of the duration of the one analysed in the current study but in the same atmospheric conditions. In this case, roughness was applied on the entire blade surface using Shin's relation.

From these results, it is clear that the research on numerical simulations of icing on wind turbines should focus on water impingement limits and roughness height. Regarding the impingement limits, better results may be obtained by considering

unsteady ice accretion simulations. However, the detail required for time discretisation is unknown. This is not sufficient, since it is not possible to obtain reliable results by using the classical empirical correlations for $k_s$ coming from the aeronautic field. These relations were developed for different systems operating in completely different environments. In-situ roughness measurements are required to remove uncertainty on this parameter. Proper numerical predictions would allow an improvement in the design of ice protection systems and wind turbine controllers during icing events.

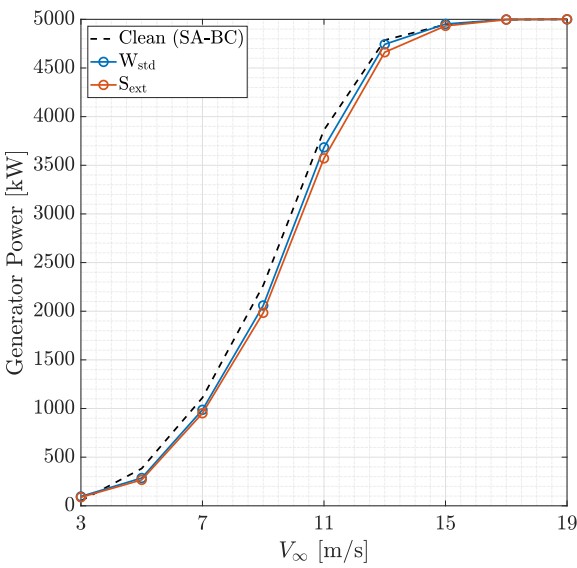

**Figure 26.** Power curve with a turbulent inflow.

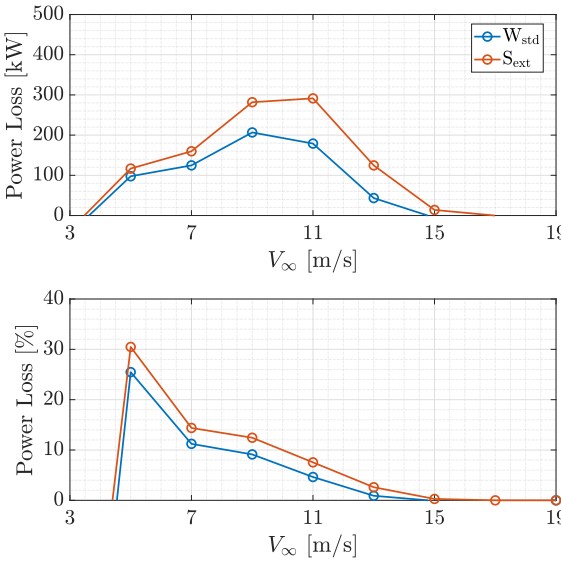

**Figure 27.** Power losses with a turbulent inflow. Top: absolute difference. Bottom: normalised difference.

# 5  Conclusions

In this paper, we conducted a detailed numerical simulation of ice accretion on the NREL 5 MW wind turbine blade using the BEM approach. To increase the precision in the computation of ice shapes, we proposed to use independent time steps during a multi-step ice accretion simulation. Moreover, we showed that it is possible to reduce the computational time required for ice accretion simulations by increasing the error of the collection efficiency and adding a very small ice thickness during each step.

Then, we analysed the effect of roughness on the aerodynamic performances of the iced sections. Due to the uncertainty of these parameters, we considered two roughness heights and two roughness extensions on each section. We computed the aerodynamic coefficients for each case and we assessed whether the aerodynamic penalty was due to ice, roughness, or both. It was shown that roughness can significantly affect the aerodynamics of an iced section, even when a complex ice shape is present, as long as $k_s$ is sufficiently high.

Finally, we computed the power curves for the low-roughness ($W_{std}$) and the high-roughness ($S_{ext}$) cases and compared them with the results of the clean wind turbine. We computed a Weibull-averaged power for each case to introduce a single figure indicating the severity of the icing event. The power loss was 50% higher for the high-roughness case.

This high variability in the prediction of power losses suggests two main areas of research for future work. The first one should be focused on the correct detection of the impingement limits of water droplets in the highly unsteady environment in which wind turbines work. The second one should be focused on the characterisation of roughness distribution and height on real wind turbine blades.

*Data availability.*  Data are available upon request from the corresponding author.

*Author contributions.*  Francesco Caccia contributed to the idea of the method, to the execution of the simulations and to the writing of the paper. Alberto Guardone contributed to the idea of the method and to the writing of the paper.

*Competing interests.*  The authors declare that no competing interests are present.

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
