# Peer review of "Numerical simulations of ice accretion on wind turbine blades: are performance losses due to ice shape or surface roughness?"

_Wind Energy Science, 2022_

## Author Comment (AC1)

Dear Reviewer,

We would like to sincerely thank you for taking the time for this in-depth review of our article. We really appreciated it. By addressing your comments, we think we have improved the technical quality of the paper.

You can find our answers below each of your comments. In *italic blue*, you can also see how we addressed the comments within the manuscript. The references we have included in our answers can be found in the revised manuscript.

Regards,

The Authors

**Anonymous Referee #1**

General comments:

The authors investigate the ice accretion on a wind turbine. The article seems original because it allows to quantify the effect of the ice surface roughness on the performance losses of the wind turbine. For this, the authors use numerical simulations. Being well aware of the strong uncertainties on the roughness input data, they performed an interesting parameterization. Moreover, they have performed quite fine simulations of the ice accretion on each studied section by a time-efficient multi-step approach.

Specific comments:

**1. Introduction**
**Reviewer Comment**
[Line 101] "The icing event was long enough for ice horns to form, to combine the effects". The author are supposed to address rime-ice conditions from an earlier comment. The term "horn" is more often used for glaze-ice shapes.
**Authors Response**
Thank you for noting.
*We have replaced "ice horns" with "streamlined, protruded ice shapes" (ll. 121-122).*

**2. Methodology**
**Reviewer Comment**
[Line 129] What is the "wind shear exponent"?
**Authors Response**
The wind shear exponent models the vertical velocity profile $V(z)$ of the wind. Specifically, it is the parameter $\alpha$ in the definition of the Normal Wind Profile model of the DNV-GL Guideline for the certification of wind turbines:

$$V(z) = V_{\text{hub}}(z/z_{\text{hub}})^{\alpha}$$

*We have defined the quantity in the text (ll. 154-157).*

**Reviewer Comment**
[Line 131] What does the OpenFAST simulation imply for the ice accretion simulation? For instance, do the wind turbine operation data account for the retroaction of the ice shape growth? (rotational velocity, etc.)
**Authors Response**
Within this work, OpenFAST computed the correct equilibrium condition of the whole system,

specifically of the blade sections, taking into account the wind shear and blade deformability. As later mentioned in the text (ll. 329-333 and ll. 428-431), we did also include a system for retroaction (which may work with any BEM method and does not explicitly require OpenFAST), based on the one proposed by Zanon et al. (2018). We decided to update the aerodynamic coefficients when the estimated difference in the AoA was higher than 0.5°. However, ice accretion was computed considering the $W_{std}$ case, and the angle of attack increased approx. by only 0.4° at the end of ice accretion. Thus, the retroaction system has never come into operation during this specific simulation, and the description of the method was omitted in the paper. It will be interesting to analyse the effect of roughness on the ice shapes themselves through this retroaction system. However, such analysis should be carried out once more accurate roughness and impingement models have been developed.

*We have provided more details where OpenFAST is first mentioned in the text together with ice accretion (ll. 158-161).*

**Reviewer Comment**

[Line 146] It seems to me a good idea to define an average power value. But why use the Weibull distribution rather than another one?

**Authors Response**

The Weibull distribution that we use, with $k = 2$ to match a Rayleigh one, is the one advised by the DNV-GL Guideline for certification of wind turbines in the standard wind turbine class.

*We have specified it in the text (ll. 190-191).*

**Reviewer Comment**

[Line 206] This is not clear to me how and why this extrapolation is performed. If it is common practices, is there any reference available?

**Authors Response**

Extrapolation is standard practice and is carried out for two reasons, i.e., to guarantee the convergence of the BEM method, which is iterative and may require data outside of the range provided without extrapolation, and to provide high-AoA aerodynamic coefficients for the root sections. We used the Viterna Method for extrapolation (Viterna and Janetzke, 1982). We also forgot to mention the correction of the 2D polars for 3D effects, which modify the behaviour in post-stall conditions. In this case we used the corrections by Du and Selig (1998) with Eggers CD adjustment (Eggers et al., 2003). These were only mentioned at the beginning of Section 2.

*We have included the references to the methods, the extrapolation tool used and the inputs in the text (ll. 146-150; 171; 249-250).*

**Reviewer Comment**

[Line 209] "the average flow field is resolved down to the Kolmogorov length scale." This seems misleading to me. This looks more like the definition of DNS simulations. The low-Re approach is related to the description of the turbulent boundary layer structure and requires y+=1 to capture the region of the viscous sublayer.

**Authors Response**

Thank you for noting.

*We have corrected the text (ll. 255-258).*

**Reviewer Comment**

[Line 236] Is there any reference for uhMesh? What kind of mesh generation technique is used?

**Authors Response**

We forgot to add the reference. The O-grid surrounding the airfoil is generated with an advancing-front technique, with the possibility of adding triangles locally. The unstructured grid around it is generated using a Delaunay triangulation, which was computed with a Bowyer-Watson algorithm.

*We have added the reference and included details on the generation algorithms when the grid was first introduced (ll. 262-263).*

**Reviewer Comment**

[Line 247] "The output had a 1P component". What does that mean?

**Authors Response**

It is a periodic oscillation with its main frequency corresponding to the rotational frequency of the rotor. The text also contained an inaccuracy since the 1P frequency is caused not only by blade flexibility, but also by the wind shear and by the tilt and cone angles of the rotor.

*We have specified the meaning of the term in the text, fixed the inaccuracy, and added quantitative data about the oscillations of $\alpha$ and $V_{\text{rel}}$ (ll. 330-332).*

**3. Validation**

**Reviewer Comment**

[Line 273] Since the description of the setup is diluted over several sections, it is not fully clear to me what experimental conditions are simulated in section 3.1.

**Authors Response**

Thank you for your comment.

*We have reported the experimental conditions and references at the beginning of the section (ll. 371-375).*

**Reviewer Comment**

[Line 298] Since the residual seems to be of importance for the methodology, it would be worth describing exactly how it is computed.

**Authors Response**

It is first worth spending some words on the iterative process to compute the collection efficiency. We used a strategy to automatically refine the seeding region by adding new particles where needed. A uniform seeding front was initialised as a linear grid with equally spaced elements. At the first iteration, the parcels not hitting the airfoil (except for the two innermost ones) were identified and removed so that the seeding front was reduced in size. The first two parcels flying just above and below the object were not removed, so the impingement limits were also refined. Then, elements were incrementally split at each iteration, evolving the current cloud front and computing the collection efficiency $\beta$ on the target surface. The simulation stopped when the difference in the $L2$ norm between two consecutive iterations of computations of $\beta$ is below a user-supplied threshold. This is what we meant by "residual" of the $k^{\text{th}}$ iteration. The term residual is indeed inaccurate. We will refer to it as "error":

$$\|\text{err}\beta\|_2 = \left( \sum_{j=1}^{n} \left[ \left( \beta_j^{[k-1]} - \beta_j^{[k]} \right) \Delta s_j \right]^2 \right)^{\frac{1}{2}}$$

*We have added a paragraph in Section 2.3 describing this process. We have defined the quantity $\|\text{err}\beta\|_2$ and replaced the misleading term "residual" accordingly (ll. 307-318; 405; 406; 413; caption of Fig. 12 and 13).*

**4. Results and discussion**

**Reviewer Comment**

[Line 327] Does the roughness always cover the whole ice surface (in the std and ext case)? On the contrary, can it cover the blade surface further than the ice?

**Authors Response**

Roughness always covers the ice surface. In the *ext* case, it covers both ice and goes 0.44m further than the end of ice impingements limits.

*For clarity, we have rewritten Section 2.4 including more details, added Figure 4 defining the std and ext cases, and added Table 3 defining the test matrix (ll. 349-369). Then, in Section 4.2, we have referenced the updated section, the table and the figure (ll. 449-450).*

**Reviewer Comment**
[Line 338] "the ice shape was mainly responsible for the aerodynamic penalty". It would be interesting to know the polar for the smooth-wall simulation of the iced shape to support this assertion.

**Authors Response**
The plots of the aerodynamic coefficients already contain five curves each and we think that adding one may lead to a lack of readability. After evaluating if adding the smooth-wall-iced coefficients in either a $C_L(C_D)$ curve or a $C_L/C_D(\alpha)$, we opted for the latter. The figure was added for Section B, being between Section A and Section C. Results for the other two sections are comparable. For completeness, we have included the results of the clean section without transition modelling in the figure, to partially address the next Reviewer comment.

*We added Figure 21 in the manuscript and modified the text accordingly (ll. 463-468).*

**Reviewer Comment**
[Line 343] "due to the supposed early transition", I do not understand this early transition. Is the transition modeled for the rough-wall simulations? If not, wouldn't it be fairer to compare against the clean simulations without transition?

**Authors Response**
Transition is not modelled for rough wall simulations. However, we believe that the assumption of fully-turbulent flow is reasonable and produces more accurate results than considering a transition model not accounting for roughness (such as the one available in SU2). In reality, the presence of roughness causes an almost instantaneous transition of the flow. However, the transition region can rather long. According to Feindt (1957), $Re_{k_s,cr} = \frac{\rho U_\infty k_s}{\mu} = 130$ is the critical $k_s$ Reynolds number for roughness to affect transition. With $Re_{k_s}$ increasing, the width of the transition region decreases, and the transition point is moved upstream. $Re_{k_s}$ on the outer half of the blade rotating at 11 rpm ranges from 750 at mid-span considering $k_s = 0.3 \cdot 10^{-3}$ to 15000 at the blade tip considering $k_s = 3 \cdot 10^{-3}$. For this reason, we think that it is correct to compare the fully-turbulent rough simulations with the simulations of the clean airfoil accounting for transition. However, this shouldn't be the case at the beginning of ice accretion. Roughness should be small and the effect of the ice shape almost negligible. For this reason, in Figure 21 we have included also the efficiency of the fully-turbulent airfoil, to add the effect of the anticipated transition in the qualitative "superposition of effects" proposed in the figure.

*We have added a comprehensive motivation of this fully-turbulent hypothesis at the end of Section 2.2 (ll. 270-287).*

**Reviewer Comment**
line 356, "This effect is peculiar since roughness should have little effect on the aerodynamic coefficients when ice horns are well developed." Is there an explanation?

**Authors Response**
The extended roughness region caused a high increase in skin friction in a geometrically smooth region, increasing the viscous drag. Moreover, the flow expansion on the suction side of the sections and compression on the pressure side were lower, causing a noticeable reduction in lift. The effects were more and more noticeable as $k_s$ increased.

*We have added this explanation in the text (ll. 487-490).*

**Reviewer Comment**
[Line 399] "Once more, our results agree with those by Etemaddar et al.", in which sense do the results agree? They may be consistent with each other but they are not in agreement (except if the figures in the table are wrong).

**Authors Response**

Thank you for noting. We wanted to highlight the consistency between the results, indeed.
*We have corrected the text (ll. 536).*

**Technical Corrections**

**Reviewer Comment**

Lines 62 and 66: 20 microns, 25 microns

**Authors Response**

We prefer to keep the metric system unit symbol for consistency with the rest of the paper.

**Reviewer Comment**

[Figure 10, page 14] Why are there systematically 2 curves for "Clean" (and the slope is not recovered)?

**Authors Response**

The two red curves represent the analytical relations for the viscous sublayer ($u^+ = y^+$) and the logarithmic region of the boundary layer $u^+ = \frac{1}{\kappa}\log y^+ + 5.1$ and were included for reference only. The label of the first subplot was wrong as well (it was supposed to be $k_s^+ = 282$).

*We have updated the figure (now Figure 11): (1) by merging the two red curves into a single one; (2) by correcting the value of $k_s^+$ on the first subplot; and (3) by replacing the lower row with results from another simulation with $k_s/c = 0.005$, i.e., one order of magnitude greater than the first row. The last point was made to be consistent with both the "W" and "S" cases analysed later. The text was modified accordingly (ll. 392-401).*

**Reviewer Comment**

[Line 370] define what TSR and Cp are

**Authors Response**

Thank you for noting that the definition was missing.

*We have added a brief description of the CP-TSR curves in the Methodology and defined the quantities (ll. 172-179).*

**Reviewer Comment**

Reference Lavoie et al (line 494): The journal article https://doi.org/10.2514/1.C036492 is probably more accessible to most readers

**Authors Response**

Thank you for noting.

*We have updated the reference (ll. 653-654).*

**Reviewer Comment**

Reference McClain et al (line 498): The journal article https://doi.org/10.4271/2019-01-1993 may also be more accessible to most readers than the conference article referenced. However, this is not exactly the same topic (although the main information that roughness evolves both in space and time is also given).

**Authors Response**

Thank you for noting.

*We have kept both references to provide the more accessible article as well (ll. 661-664).*

---

## Author Comment (AC2)

Dear Reviewer,

Thank you for taking time to review our article. With your suggestions you have provided valuable insight on how to improve the paper. As suggested, we have added more validation test cases for both ice accretion and roughness. We have also made the paper clearer by adding significant images and tables.

You can find our answers below each of your comments. In *italic blue*, you can also find how we addressed the comment within the manuscript. The references included in our answers can be found in the revised manuscript.

Regards,

The Authors

**Anonymous Referee #2**

This paper investigated the effects of roughness on airfoil performances under ice conditions. In this paper, NERL 5MW model was applied to perform numerical simulations. To analyse the power performance of the iced blade, DLC1.1 was considered with two different roughness cases, W_std and S_std. Detailed comments are addressed below.

Comments
**Reviewer Comment**
IEA task 19 published a report about available technologies for wind energy in cold climates where detailed IPS cases were reported. This report should be reviewed
**Authors Response**
Thank you for your suggestion. We have reviewed the report. We focused on ice accretion models rather than IPS, the former being more in line with our paper.
*We have updated the text and included a review of the report in the Introduction (ll. 34-45).*

**Reviewer Comment**
Recently, there were studies to investigate surface roughness effects for simulating ice accretion. These papers should be reviewed.
(https://arc.aiaa.org/doi/10.2514/1.J060641, https://arc.aiaa.org/doi/10.2514/1.J059222)
**Authors Response**
Thank you for your suggestion. The first paper deals with boundary conditions in roughness-induced transition with no explicit relation with ice accretion simulations. The second paper includes a roughness-induced transition in ice accretion simulation, showing that transition is important for glaze ice shapes (as it affects heat transfer), while it is negligible for rime ice simulation. We have included the latter in the methodology to support neglecting transition in rime ice simulations. Feindt's experiment on the roughened flat plate (Feindt, 1957), which is used for validation in both papers, was used to explain why using fully-turbulent flows provides an acceptable approximation when computing aerodynamic coefficients of iced airfoils.
*The papers were reviewed at the end of Section 2.2 (ll. 270-287).*

**Reviewer Comment**
In section 2.4 the authors introduced the extended roughness area: 25%, 18%, 15%, 13%, and 11% along the blade span shown in Fig. 3, respectively. I think it is one of the most important parts of this paper. But there is no detailed description of how these values were introduced. It must be clearly described.
**Authors Response**
The extended roughness region is 0.44m and is constant among all sections, so its non-dimensional

value (non-dimensionalized with respect to the chord of the section) reduces from tip to root. At the tip, the value is 25%. At mid-span, the value is 11%. The value of 25% at the tip was chosen to match the one imposed by Etemaddar et al. (2014) on all sections. In this study, it was chosen not to keep this value constant in non-dimensional value, since it would be physically wrong. Larger sections generate higher pressure gradients, which deviate more the droplets. So, they collect fewer water droplets as compared the smaller sections at the tip. For this reason, the rough region should be higher at the tip and diminishing going towards the root.

*We have clearly explained this concept within the text (ll. 349-369). For clarity, Figure 4 representing the "std" and "ext" regions was also included, as well as Table 3 with a test matrix.*

**Reviewer Comment**

In Figs. 11 and 12 two different comparison studies were presented. It was shown that the current numerical results were not able to accurately predict the ice shape compared to the experimental test results. The author mentioned that "the ice impingement limit on the lower surface was underestimated". It might be due to that impinged water does not freeze at the surface and exists as a water film. Therefore, it might be good to check the heat transfer rate.

**Authors Response**

The ice shape on the leading edge was actually captured fairly well with the proposed methodology. We updated the figure by increasing the sampling rate on the experimental data. The test case is fully rime and there is no runback water (there are both a low freestream temperature and a very low LWC). The most likely reason for the underestimation of the impingement limit with respect to the experiments (which is common to any numerical ice accretion engine) is that a cloud is made by a distribution of droplet diameters and the MVD is just an indicator of the median of this distribution. Parcels with higher diameter have a higher mass, and their trajectory is less deflected by the pressure gradient. Thus, a wider portion of the airfoil gets wet. This issue may be overcome with a multi-bin approach, i.e., by performing the weighted average of the collection efficiency computed with different droplet diameters from the distribution. However, this wouldn't affect significantly the results and the conclusions presented, and the multi-bin approach goes beyond the scope of this work.

To prove that our results are consistent with experiments and other ice accretion engines, the six remaining AERTS test cases for rime ice were included, i.e., #4, #15, #16, #17, #18, and #19. For these test cases, a LEWICE solution is available as well. The results obtained with the proposed multi-step ice accretion setup agreed well both with the experiments and also with those provided by Han et al. (2012) for LEWICE. Both PoliDrop-PoliMIce and LEWICE showed the same behaviour in terms of impingement limits, and the reason is the one described above.

*Section 3.2 was updated to include the six new validation test cases. Results are shown in Figure 13. The underestimation of the impingement limits on the lower surface was also explained (ll. 408-432).*

**Reviewer Comment**

In Fig. 13, how do the authors ensure the predicted ice shapes are correct? Moreover, the ice shape at the blade tip areas (section A-C) seems very irregular horn shapes. Is it obtained under the rime ice condition? In general, more validation studies are required to prove the current numerical model's accuracy.

**Authors Response**

More validation cases have been added to ensure that the ice shapes are correct. The ice shapes on all sections (A-E) are rime ice. The irregularities on Sections A-C come from some small oscillations on the collection efficiency, which eventually get amplified. This happens because geometries are not being smoothed during grid generation, unless strictly required. Since real ice shapes are highly irregular, this was considered acceptable.

*More validation test cases have been added (Figure 13). We have also specified that the geometry is not subject to smoothing, unless strictly required for grid generation purposes (ll.304-305; 442-434).*

**Reviewer Comment**

On page 20, section 4.3, why is after the optimum TSR value interested?

**Authors Response**

High TSR values are used by this wind turbine in Region 1.5, i.e., from the cut-in wind speed (3 m/s) up to approx. 8 m/s. In this velocity range, the TSR decreases from 15.3 to its optimum value.

*We have added this information in the text by specifying what is the Region 1.5 (ll. 506-507).*

**Reviewer Comment**

Figure 23 shows the power curve under turbulent wind conditions. Why clean airfoil case where no ice is accumulated could not obtain the rated power, 5MW, at the rated wind speed? Since there is no ice accumulated with the clean airfoil model, it should produce the rated power at the rated wind speed.

**Authors Response**

The rated power is produced at the rated speed with a steady wind, i.e., when the turbulence intensity (T.I.) is zero. As the T.I. increases, the 10-minute averaged power reaches the rated power at higher wind speeds. This information can be found for instance in https://doi.org/10.1088/1742-6596/524/1/012109 from which picture below was taken.

[Figure]

The concept can be shown with some reasoning. Let us consider a simplified example. Given an average wind speed of 13 m/s, which is above rated speed, the instantaneous one will be, e.g., 13 m/s $\pm$ 4 m/s. At instantaneous wind speed above rated one, the wind turbine will produce approximately the rated power. At instantaneous wind speed below the rated one, the wind turbine will produce a below-rated power. Thus, the average power will be necessarily below the nominal rated power. For similar reasoning, before the "inflection point", the power produced is slightly higher than the 0 T.I. case.

*We have specified the effects of turbulence intensity in the text (ll. 518-520). For completeness, we have clearly specified the reference T.I. prescribed by DLC 1.1 for the wind turbine under analysis (l. 184).*

**Reviewer Comment**

Overall, this paper needs more validation studies to prove that the current model is valid for 3D wind turbine rotor simulations. Furthermore, roughness model validations are required.

**Authors Response**

A classic quasi-3D BEM approach was used for icing. It is known that the model is valid for 3D simulations, in the sense that it is possible to approximate 3D ice accretion simulations with 2D sections (Switchenko et al., 2014). The validation of our numerical setup for ice accretion has now been expanded including new test cases.

The validation of the roughness model was already shown in Section 3.1 by comparing the law of the wall on a rough airfoil with the theoretical velocity profile. Since we considered only $k_s/c = 0.0005$, we have included the results of another simulation with $k_s/c = 0.005$. The figure has been modified so that in the first row there are the results of $k_s/c = 0.0005$ on the upper surface of the airfoil, while in the second row there are the results of $k_s/c = 0.005$ on the lower surface of the airfoil. The results of the law of the wall on a smooth airfoil were also presented and compared with the theoretical velocity

profile.

*We have updated Section 3.1 by adding another value of $k_s/c = 0.005$ for the validation of the roughness model. (ll. 392-401 and Figure 11).*

**Reviewer Comment**

Many studies have already investigated the power performance of a wind turbine with and without ice accumulations. Therefore, there is no novelty in evaluating the power performance with a CFD tool. One of the most exciting parts of this paper is considering the surface roughness effects. However, it is not clear how different surface roughness was considered and implemented into the simulations.

**Authors Response**

We have updated the description of how the different roughness cases were chosen, as specified in a previous answer. We have also slightly expanded the description of the law of the wall, including the smooth regime. The description of the roughness-modified Spalart Allmaras turbulence model and its numerical implementation in SU2 is available in the work by Ravishankara et al (2020). The turbulence model itself is presented in the work by Aupoix and Spalart (2003). Presenting this turbulence model and its numerical implementation is out of the scope of this paper.

*We have included the description of the law of the wall in the smooth regime (ll. 210-214). The reference (Aupoix and Spalart, 2003) was missing and has been added in the text (ll. 252, 573-574.)*

**Reviewer Comment**

Based on the aforementioned comments, this reviewer recommends rejecting this paper.

---

## Author Response (AR2)

Dear Reviewer,

We would like to thank you again for reviewing the updated version of our manuscript and for the comments provided.

You can find our answers below each of your comments. In *italic blue*, you can find how we addressed the comment within the manuscript. The references included in our answers can be found in the revised manuscript.

Regards,

The Authors
* * *
I thank the authors for the improvements made in their article. The new version is OK for me. I just have the following few comments:

**Reviewer Comment**
Regarding the new equation (1) added in order to define the wind shear exponent, is z defined anywhere? and V? V is the horizontal velocity rather than the vertical velocity?
**Authors Response**
The coordinate $z=0$ is defined at ground level, while V is the average horizontal wind speed.
*We have included the definition of V and z in the text (l. 154).*

**Reviewer Comment**
Regarding the comment "Lines 62 and 66: 20 microns, 25 microns", my concern was not that the unit is written microns or $\mu m$ but the values. 0.2 and 0.25 microns seem small to me.
**Authors Response**
Thank you for clarifying it. As you state, the correct values are 20 microns and 25 microns.
*We have updated the values in the manuscript (ll. 82 and 87).*

**Reviewer Comment**
In the new paragraph from line 34 to line 45, it would be better to cite more recent codes or references, like the ones related to the AIAA-IPW1. Especially for FENSAP-ICE: https://doi.org/10.2514/6.2022-3311 and ONERA's codes: https://doi.org/10.2514/6.2022-3310. Maybe add GlennIce too for NASA's codes: https://doi.org/10.2514/6.2022-3309.
**Authors Response**
Thank you for the suggestion. The references were taken from the report we were reviewing and have now been updated.
*We have updated the manuscript by referring directly to the IPW1, adding the suggested references, our contribution to the IPW1 (Morelli et al., 2022), and the review of all the results (Laurendeau et al., 2022) (ll. 44-47).*

**Reviewer Comment**
line 206: Von K\'arm\'an
**Authors Response**
Thank you.
*We have updated the text (l. 205).*

**Anonymous Referee #3**

Dear Reviewer,

Thank you for taking time to read through the previous review iteration and for reviewing our manuscript.

You can find our answers below each of your comments. In *italic blue*, you can also find how we addressed the comment within the manuscript. The references included in our answers can be found in the revised manuscript.

Regards,

The Authors
* * *
After reading through the paper and comparing the previous reviews against the modified version of the paper, I have only few comments to add.

**Reviewer Comment**
In the intro it is stated that wind energy represents 16% of the total energy mix. This is not correct; It is only 16% of the electricity production and only 3-4% of the total energy production in Europe.
**Authors Response**
Thank you for the correction.
*We have corrected the text and included more recent data about 2022, which state that electricity production was 15% (Jones et al., 2023) (ll. 13-14).*

**Reviewer Comment**
In the intro, it is also stated that higher wind speeds and air density guarantees a higher wind power density in cold regions, and it is essentially the creation ice that lowers down the power production. However, it should also be mentioned that cold climate often leads to stable ABL with less mixing, which subsequently means stronger wake effects, and a reduction of the power production in wind farms. So, it is actually not only ice effects that lower the energy production.
**Authors Response**
Thank you for this comment.
*We have updated the text to include this information (ll. 17-20).*

**Reviewer Comment**
The geometry of the horn seen in the ice accretion in Figs. 14 and 15 looks very complicated and must be very challenging for the grid generator to capture. I suggest that the grid associated with these figures are also shown in order to demonstrate the quality of the grid.
**Authors Response**
It is indeed challenging to generate the grid in highly concave regions. To retain grid quality, uhMesh adopts a hybrid advancing front technique to generate a boundary layer in which triangles are added in concave/convex region. For the computation of the aerodynamic coefficients, grid quality was further improved by reducing the extent of the concave regions in Sections A and B. This didn't affect the overall ice shape but allowed faster reduction of the residuals of the simulations at all the angles of attack. This information was not included in the original manuscript. Thus, it has been added.
*We have updated Fig. 15 by superimposing the computational grid used for the aerodynamic coefficients onto the ice shape coming from the ice accretion simulations. A red box was also added to highlight the region shown in the new Fig. 16, where the orthogonality of the grid is presented. The text was updated to introduce these figures and the modification of the concave regions (ll. 437-442).*